

# Exploring ship track spreading rates with a physics-informed Langevin particle parameterization

Lucas A. McMichael [1], Michael J. Schmidt [2], Robert Wood [1], Peter N. Blossey [1], and Lekha Patel [2]

[1]University of Washington, Seattle, WA, USA
[2]Sandia National Laboratories, Albuquerque, NM, USA

**Correspondence:** Lucas A. McMichael (mcmic@uw.edu)

**Abstract.** The rate at which aerosols spread from a point source injection, such as from a ship or other stationary pollution source, is critical for accurately representing subgrid plume spreading in a climate model. Such climate model results will guide future decisions regarding the feasibility and application of large-scale intentional marine cloud brightening (MCB). Prior modeling studies have shown that the rate at which ship plumes spread may be strongly dependent on meteorological conditions,

such as precipitating versus non-precipitating boundary layers and shear. In this study, we apply a Lagrangian particle model (PM-ABL v1.0), governed by a Langevin stochastic differential equation, to create a simplified framework for predicting the rate of spreading from a ship-injected aerosol plume in sheared, precipitating, and non-precipitating boundary layers. The velocity and position of each stochastic particle is predicted with the acceleration of each particle being driven by the turbulent kinetic energy, dissipation rate, momentum variance, and mean wind. These inputs to the stochastic particle-velocity equation

are derived from high-fidelity large-eddy simulations (LES) equipped with a prognostic aerosol-cloud microphysics scheme (UW-SAM) to simulate an aerosol injection from a ship into a cloud-topped marine boundary layer. The resulting spreading rate from the reduced-order stochastic model is then compared to the spreading rate in the LES. The stochastic particle-velocity representation is shown to reasonably reproduce spreading rates in sheared, precipitating, and non-precipitating cases using domain-averaged turbulent statistics from the LES.

## 1   Introduction

Lying beneath regions of large-scale subsidence and above cool subtropical eastern ocean basins, stratocumulus clouds exert a strong negative radiative forcing (cooling effect) on the climate as these bright low clouds are able to efficiently reflect shortwave radiation back to space in comparison to the relatively dark ocean surface below (Hartmann and Short, 1980). Stratocumulus decks observed from space often reveal narrow bands of enhanced albedo (known as ship tracks) as a result of

ship emissions (Conover, 1966). The underlying physical mechanism behind the observed cloud brightening in ship tracks is referred to as the Twomey effect, which describes the relationship between increasing the number of cloud condensation nuclei (CCN) and the corresponding higher albedo as the increased surface area of the resulting smaller, more numerous droplets act as a more reflective surface than a lower CCN environment (Twomey, 1974, 1977). However, the Twomey effect does





not act in isolation and various additional aerosol-cloud interactions may occur with an increase of CCN, such as reduced

collision-coalescence efficiency and the suppression of precipitation (Albrecht, 1989).

Because the Twomey effect could be enhanced by increased liquid water path (LWP) and/or extended cloud lifetime associated with precipitation suppression, Latham (1990) postulated that the injection of sea salt aerosols into stratocumulus-topped boundary layers (Marine Cloud Brightening; MCB) may be a viable method to offset a substantial portion of greenhouse warming and potentially circumvent climate tipping points, such as a collapse of the arctic ice sheet (Rasch et al., 2009).

However, the clouds do not always increase in brightness when aerosol concentrations are increased. For example, aerosol increases in non-precipitating clouds can lead to decreases in cloud fraction and liquid water path (Ackerman et al., 2004; Toll et al., 2017). In models, the smaller droplets (that result from the increase in CCN with added aerosols) more readily evaporate (Wang et al., 2003), in part because they remain near cloud top for longer periods of time as a result of decreased sedimentation rates (Ackerman et al., 2004; Bretherton et al., 2007). The aforementioned properties of smaller droplets promote increased

entrainment efficiency, which may decrease LWP/cloud fraction and act to darken the cloud (Wood, 2007). Observational estimates of the degree to which an aerosol perturbation may act to alter LWP shed little light on the aerosol-cloud response, with studies finding depleted LWP (Christensen et al., 2023; Sato et al., 2018; Diamond and Wood, 2020; Gryspeerdt et al., 2019; Segrin et al., 2007; Coakley and Walsh, 2002), augmented LWP (Gryspeerdt et al., 2019; McCoy et al., 2018), and background meteorological condition dependence (Bender and Sentelhas, 2018; Christensen and Stephens, 2011, 2012).

Given the muddled results from satellite retrievals and the grid spacing required to numerically resolve both ship tracks and mesoscale circulations, process studies regarding the response of low clouds to aerosol injections often employ large-eddy simulation (LES), capable of resolving fine-scale turbulent structures that are crucial for estimating local aerosol/microphysical process rates and cloud-top entrainment rates (Lewellen and Lewellen, 1998). Previous LES studies of ship tracks have shown the LWP response to be dependent on background aerosol concentrations, with clean boundary layers exhibiting larger LWP

in the ship track region and polluted boundary layers experiencing the opposite (Wang et al., 2011; Berner et al., 2015; Chun et al., 2023). In a LES study of an idealized summertime subtropical stratocumulus regime, Chun et al. (2023) found that regardless of background aerosol or free tropospheric moisture, the Twomey effect remained larger than the cloud fraction and LWP adjustments in all cases and resulted in cloud brightening of varying magnitudes over a 2-day period. Considering these promising LES results and a greater understanding of the environments in which MCB would be most effective, global

modeling efforts of aerosol injections are a critical component in determining the feasibility of large-scale deployment and illuminating potential downsides related to regional climate variability brought about by inhomogeneous radiative forcing (Latham et al., 2008; Jones et al., 2009; Rasch et al., 2009).

Global climate model (GCM) studies of regional aerosol/CCN perturbations of the eastern subtropical oceans have suggested that MCB may be able to offset much of, if not all, the warming from a doubling of $CO_2$ (Jones et al., 2009; Rasch et al., 2009;

Hill and Ming, 2012; Ahlm et al., 2017). However, the nature of the regional and global climate responses may differ substantially across the perturbation strategies and the assumed size distributions of aerosols (Wood, 2021). One major complicating factor of using GCMs to probe MCB feasibility is that they suffer from insufficient low cloud cover over the eastern subtropical ocean regions in comparison to observations (Xie et al., 2018), primarily as a result of under-resolved vertical and horizontal





processes (Lee et al., 2022). Global-scale high-resolution (2-5 km horizontal grid spacing) simulations have recently become
possible; however, such configurations are only practical for the simulation of a few days to months (Khairoutdinov et al.,
2022), and not the decades or centuries that are required of GCMs. Future GCM runs of MCB strategies that do not involve
instantaneous perturbations of entire ocean basins will require information about how injected aerosol plumes spread within the
grid, and such information will need to be relayed to the radiation and microphysics parameterizations to better capture the spa-
tial heterogeneity at small scales. In addition to MCB, a computationally efficient model of aerosol spreading tied to turbulent
dynamics may be broadly applicable to injected stratospheric aerosols and the spread of hazardous chemicals/aerosols.

Previous attempts to constrain ship track spreading rates from satellite images aimed to estimate average lateral plume
spreading behavior (Durkee et al., 2000; Patel and Shand, 2022), but assuming a constant rate of plume spreading may result
in subgrid plume-fraction imprecision that leads to compounding, non-linear errors on the resolved-scale properties as prior
LES modeling studies have shown that the rate at which ship plumes spread may be strongly dependent on meteorological
conditions, such as precipitating versus non-precipitating boundary layers (Prabhakaran et al., 2023), or wind shear (Berner
et al., 2015). Recent efforts to represent subgrid plumes, such as the Plume-in-grid (PIG) method with adaptive grids (Sun
et al., 2022), allow for time-dependent changes in the horizontal spreading rate as a function of wind shear but require grid
refinements in the presence of plumes. The associated increase in computational demand may be a bottleneck for the assessment
of large-scale injection strategies. Alternatively, by leveraging Lagrangian particle-based methods, subgrid plumes may be
characterized by statistical descriptions of the flow field at singular points in space and time and not bound by standard Gaussian
diffusion/dispersion that assumes fixed plume dispersion rates (Pope, 2000). The utilization of scalable approaches necessitates
an accurate and computationally efficient representation of subgrid particle trajectories.

In this study, we formulate a turbulence-driven Lagrangian particle model, governed by a Langevin stochastic differential
equation, to create a simplified and computationally efficient framework for predicting the rate of horizontal spreading from
a ship-injected aerosol plume. The velocity and position of each stochastic particle is controlled by the acceleration of the
surrounding fluid, which depends on the turbulent kinetic energy (TKE), dissipation rate, momentum variance, and mean wind.
These inputs to the stochastic particle-velocity equation are derived from high-fidelity, large-domain (204.8 km × 25.6 km)
large-eddy simulations (LES) equipped with a prognostic aerosol-cloud microphysics scheme (Berner et al., 2013) to simulate
aerosol injection from a ship into a cloud-topped marine boundary layer. By minimizing the error between the Gaussian fits of
the Lagrangian particle model and LES ship track widths, we constrain a free parameter within the particle model established
in Pope (2000). Using the fully-parameterized particle model, we then study horizontal ship track spreading rates across a
range of plausible vertical shear magnitudes in the northeast Pacific boundary layer under precipitating and non-precipitating
conditions. Conditionally-averaged turbulent statistics both from within the ship plume and across the entire domain are used
to judge spreading rate sensitivity and behavior in the particle model.

The paper is organized as follows: Section 2 describes the theoretical background of the Langevin particle model and
details the numerical implementation of the approach. Section 3 outlines the LES configurations, Langevin particle model
input parameters, and the plume width calculation used to compare the LES to the particle model. Section 4 compares the



LES and particle model plume widths across the different shear cases and discusses the input parameters needed for optimal performance. Section 5 summarizes and discusses the findings of this work.

## 2 Description of Plume Model

This work aims to efficiently model the turbulent dispersion of atmospheric aerosols in the marine boundary layer for potential use as a subgrid plume parameterization in a GCM. In service of that effort, we consider a simplified but appropriately parameterized Langevin model and employ notation that is largely based on that used in Pope (2000). Specifically, we consider the unbounded, $d = 1, 2, 3$-dimensional, turbulent dispersion of a passive, conserved (no sources or sinks) scalar tracer, $\phi(\boldsymbol{x}, t)$ with arbitrary units (e.g., temperature, concentration, etc.) and known initial condition, $\phi_0(\boldsymbol{x})$. Being *passive*, this tracer has no effect on the material properties of the air or flow field through which it is transported (i.e., density, kinematic viscosity, molecular diffusion–$\rho, \nu, \Gamma$) and thus has no effect on its own transport mechanism.

In the following sections, we will lay out the equations that govern our atmospheric plume model. We will begin with the Eulerian formulation and from there, work towards the Lagrangian formulation that corresponds to the numerical particle model we introduce in Section 2.3.

### 2.1 Governing Equations

The Eulerian conservation equation that governs the evolution of the scalar tracer field is the advection-diffusion equation

$$\frac{\partial \phi}{\partial t} + \nabla \cdot (\boldsymbol{U} \phi) = \Gamma \nabla^2 \phi, \tag{1}$$

with the initial condition at $t_0 = 0$

$$\phi(\boldsymbol{x}, t_0) = \phi_0(\boldsymbol{x}). \tag{2}$$

The unbounded domain assumption along with the diffusive nature of the spreading process implies the following decay condition and long-time asymptotic solution

$$\lim_{x_i \to \infty} \phi(\boldsymbol{x}, t) = 0, \quad t \geq 0, \quad i \in \{1, 2, 3\}, \tag{3}$$

$$\lim_{x_i \to -\infty} \phi(\boldsymbol{x}, t) = 0, \quad t \geq 0, \quad i \in \{1, 2\}, \tag{4}$$

$$\lim_{x_3 \to 0} \phi(\boldsymbol{x}, t) = f(\boldsymbol{y}, t), \quad t \geq 0, \quad \boldsymbol{y} \in \mathbb{R}^2 \tag{5}$$

$$\lim_{t \to \infty} \phi(\boldsymbol{x}, t) = 0, \quad \boldsymbol{x} \in \mathbb{R}^3. \tag{6}$$

Here, $\boldsymbol{U}(\boldsymbol{x}, t) \in \mathbb{R}^3 \; [\text{ms}^{-1}]$ is the fluid velocity, $\Gamma \in \mathbb{R} \; [\text{m}^2\text{s}^{-1}]$ is the constant diffusion coefficient (exact formulation depending on the definition of $\phi$), and $f(\boldsymbol{y}, t)$ is an arbitrary boundary condition at the earth's surface ($x_3 = 0$). Note that we formulate this for $d = 3$, though, formulations for $d = 1, 2$ may be written similarly.



The evolution of the fluid velocity field, $\boldsymbol{U}(\boldsymbol{x},t)$ is governed by the Navier-Stokes equations

$$\frac{\mathrm{D}\boldsymbol{U}}{\mathrm{D}t} = -\frac{1}{\rho_0}\nabla p + \nu\nabla^2\boldsymbol{U} + F_s, \tag{7}$$

where we make the assumption that the base-state fluid density only varies in the vertical, $\rho_0(z)\ \left[\mathrm{kg\ m^{-3}}\right]$ (anelastic approximation). Within (7), the kinematic viscosity is defined as $\nu := \mu/\rho\ \left[\mathrm{m^2\ s^{-1}}\right]$, wherein $\mu$ is the coefficient of viscosity that is assumed to be a constant in a Newtonian fluid. The mean fluid pressure is denoted by $p\ \left[\mathrm{kg\ m^{-1}s^{-2}}\right]$ and any additional

sources or sinks of momentum (e.g., apparent forces such as Coriolis, centrifugal, or the buoyancy contributions in the vertical velocity component) are captured by the forcing term, $F_s$. Finally, the $\mathrm{D}/\mathrm{D}t$ operator is the material derivative defined as

$$\frac{\mathrm{D}}{\mathrm{D}t} := \frac{\partial}{\partial t} + \boldsymbol{U} \cdot \nabla. \tag{8}$$

By introducing a filtering operation which separates resolved fluid motion ($U_i$) from unresolved, subgrid motion ($u_i$), (7) becomes the Reynolds-averaged Navier Stokes (RANS) equation

$$\frac{\partial U_i}{\partial t} + U_j\frac{\partial U_i}{\partial x_j} = -\frac{1}{\rho}\frac{\partial p}{\partial x_i} + \nu\frac{\partial^2 U_i}{\partial x_j\partial x_j} - \frac{\partial \tau_{ij}}{\partial x_j} + F_s, \tag{9}$$

where the subgrid Reynolds stresses, $\tau_{ij} := \overline{u_iu_j}\ \left[\mathrm{m^2s^{-2}}\right]$, represent a sink of momentum brought about by unresolved (subgrid-scale) fluctuations in velocity. A turbulence closure for $\tau_{ij}$ is necessary to close the system of equations given in (9). Once a relationship between the mean flow and subgrid flow has been established and an equation for pressure and conservation of energy have been solved, the mean fluid velocity can be updated in time and space, but as is the case with LES, this

requires substantial computing power at fine spatial and temporal resolution and we instead wish to describe the flow field that dictates tracer transport in a simplified manner.

## 2.2   Lagrangian governing equations

Given a turbulent flow field, defined in terms of its mean velocity, $\overline{\boldsymbol{U}}(\boldsymbol{\mathcal{X}},t)$, over the region $\boldsymbol{\mathcal{X}} \subset \mathbb{R}^3$, Reynolds stresses $\tau_{ij}$, and scalar dissipation rate $\epsilon\ \left[\mathrm{m^2s^{-3}}\right]$, we are concerned with modeling the mean field of our aerosol plume, $\overline{\phi}(\boldsymbol{x},t)$, based on its

initial condition $\phi_0(\boldsymbol{x})$. The Lagrangian nature of our model indicates that rather than describing our system with continuous fields in time and space, we will instead follow parcels of fluid that are characterized by their position and velocity, and we deploy the notation $\boldsymbol{X}^{(n)}(t)$, $\boldsymbol{U}^{(n)}(t)$, $n = 1,\ldots,N_p$. Under this notational convention, $N_p$ is the number of particles in the system and each carries an equal portion ($m_p$) of the total mass of fluid, $\mathcal{M}$, contained in a fixed volume $\mathcal{V}$ such that

$$m_p := \frac{\mathcal{M}}{N_p} \equiv \frac{\rho\mathcal{V}}{N_p}. \tag{10}$$

We note that the assumption of equal mass for all particles is convenient but not required, as many Lagrangian models allow for unequal particle masses that may also vary in time (Monaghan, 2012; Tartakovsky et al., 2016; Avesani et al., 2015; Cherfils et al., 2012; Bosler et al., 2017; Schmidt et al., 2020, 2019; Engdahl et al., 2019; Jiao et al., 2022).



| Particle Moment | | | Fluid Property |
|---|---|---|---|
| First | $\overline{\boldsymbol{U}^*|\boldsymbol{x}}$ | $\equiv$ $\overline{\boldsymbol{U}}(\boldsymbol{x},t)$ | Mean Velocity |
| Second | $\overline{\boldsymbol{u}_i^*\boldsymbol{u}_j^*|\boldsymbol{x}}$ | $\equiv$ $\overline{\boldsymbol{u}_i\boldsymbol{u}_j}$ | Reynolds Stresses |

**Table 1.**

The particles in this model are all independently and identically distributed–i.e., two particles beginning at the same position and under the same fluid conditions (or at identical times) will be governed by equivalent position and velocity densities and thus have the same underlying probability density function (PDF). Further, their corresponding trajectories do not depend on one another. For this reason, we only need consider the dynamics of a single, arbitrary particle to describe this model, and we will denote its properties as $\boldsymbol{X}^*(t)$, $\boldsymbol{U}^*(t)$. If we accept the argument of GI Taylor (1921) that for the considered problem of atmospheric transport with a relatively high Reynolds number (Re),the molecular diffusion provides a negligible contribution to the transport of the aerosol plume, as compared to the mean fluid flow and turbulent motions. In the case where molecular diffusion is neglected, (6) is no longer valid given that the plume will fail to spread indefinitely in a flow with no turbulence. As such, the tracer $\phi$ is conserved along the path of a fluid parcel (particle), and the evolution of the mean aerosol plume field $\overline{\phi}$ is fully specified by the statistical properties of the fluid particles in motion. It is then convenient that the particle velocity PDF $f^*(\boldsymbol{\mathcal{U}}|\boldsymbol{x};t)$ is equivalent to the fluid velocity PDF $f(\boldsymbol{\mathcal{U}};\boldsymbol{x},t)$. Note here that $\boldsymbol{\mathcal{U}}$ is the independent, or dummy, variable for the PDF of the velocity random variable, $\boldsymbol{U}^*$–e.g., employing the notation more common to probability theory, the PDF could also be written as $f_{\boldsymbol{U}^*}(\boldsymbol{u}^*|\boldsymbol{x};t)$. Also, note that $f^*$ is a density for the velocity $\boldsymbol{U}^*$ conditioned on the particle being located at position $\boldsymbol{x}$, which is also a function of the time, $t$, at which the velocity is sampled; whereas, $f$ is a density for $\boldsymbol{U}$ but is not conditional and is strictly a function of $\boldsymbol{x}$ and $t$. This directly implies that the first and second moments of the particle velocity are equal to the mean fluid velocity and Reynolds stresses, as given in Table 1.

Due to the properties presented above, as well as other correspondences between the fluid and particle systems that are given in (Pope, 2000, Table 12.1), we obtain the governing equations for the particles. The position of a particle evolves according to

$$\frac{\mathrm{d}\boldsymbol{X}^*(t)}{\mathrm{d}t} = \boldsymbol{U}^*(t), \tag{11}$$

with the particle velocity obeying an Ornstein-Uhlenbeck (diffusion) process defined by the stochastic differential equation (SDE)

$$\mathrm{d}\boldsymbol{U}^*(t) = \boldsymbol{a}\left(U^*(t), \boldsymbol{X}^*(t), t\right)\mathrm{d}t + b\left(\boldsymbol{X}^*(t), t\right)\mathrm{d}\boldsymbol{W}(t). \tag{12}$$

Here, $\boldsymbol{a}\left(\boldsymbol{\mathcal{U}}, \boldsymbol{x}, t\right)$ and $b\left(\boldsymbol{x}, t\right)$ are generically formulated drift and diffusion functions, respectively, and $\boldsymbol{W}(t)$ denotes the $d$-dimensional (standard) Brownian motion. The Fokker-Planck equation associated with (11) and (12) governing the behavior of the particle velocity-position joint PDF, $f_P^*(\boldsymbol{\mathcal{U}}, \boldsymbol{x}; t)$ is

$$\frac{\partial f_P^*}{\partial t} + \mathcal{U}_i \frac{\partial f_P^*}{\partial x_i} = -\frac{\partial}{\partial \mathcal{U}_i}\left[f_P^* a_i\left(\mathcal{U}, \boldsymbol{x}, t\right)\right] + \frac{\left[b\left(\boldsymbol{x}, t\right)\right]^2}{2}\frac{\partial^2 f_P^*}{\partial \mathcal{U}_i \partial \mathcal{U}_i}. \tag{13}$$





Note than in (13) and going forward, we apply Einstein notation to represent sums over the spatial coordinate indices $\{i, j, k\}$. Finally, after traversing a small jungle of mathematical manipulations (Pope, 2000), we arrive at the *generalized Langevin model* (GLM) for the particle-velocity stochastic process

$$\mathrm{d}U_i^*(t) = -\frac{1}{\rho}\frac{\partial P}{\partial x_i}\mathrm{d}t + \boldsymbol{G}_{ij}\left[U_j^*(t) - \overline{U_j^*|\boldsymbol{X}^*(t)}\right]\mathrm{d}t + \sqrt{C_0\epsilon(\boldsymbol{X}^*(t),t)}\,\mathrm{d}W_i(t). \tag{14}$$

This SDE contains some previously-undefined terms, including the particle-pressure field $P(\boldsymbol{x}, t)$ that directly corresponds to

the mean fluid pressure $\overline{p}(\boldsymbol{x}, t)$. Additionally, we introduce the constant $C_0$ and the drift coefficient $\boldsymbol{G}_{ij}(\boldsymbol{X}^*(t), t)$, where $\boldsymbol{G}_{ij}$ depends on local values of Reynolds stresses $(\overline{u_i u_j})$, $\epsilon$ (a function of $\boldsymbol{X}^*$ and time), and cross-derivatives of the mean velocity $(\partial\overline{U_i}/\partial x_j)$. The reason (14) is referred to as generalized is because it defines a class of models that are specified according to choices for $\boldsymbol{G}_{ij}$ and $C_0$.

### 2.2.1   Simplified Langevin model

There are some attractive properties of boundary-layer flows that allow us to eliminate terms from the GLM and arrive at a modification of what is referred to as the *simplified Langevin model* (Pope, 2000). First we assume that, in the boundary layer, the large-scale (mean) pressure gradients are small enough to be negligible, i.e.,

$$\frac{\partial P}{\partial \boldsymbol{x}_i} \equiv 0. \tag{15}$$

Second, the nature of boundary layer flows admits an isotropic weight coefficient for the drift term, namely

$$\boldsymbol{G}_{ij} := -\frac{3}{4}C_0\frac{\epsilon}{k}\delta_{ij}, \tag{16}$$

where $\epsilon$ and $k \;\left[\mathrm{m^2 s^{-2}}\right]$ are, respectively, the scalar dissipation rate and turbulent kinetic energy (TKE).

Equivalently, (16) can be formulated as a constraint on the evolution of kinetic energy in homogeneous turbulence,

$$\frac{3}{2}C_0\epsilon + \boldsymbol{G}_{ij}\left(\overline{u_i u_j}\right) = 0. \tag{17}$$

This constraint also serves to define the Lagrangian integral timescale, $T_L$, that can be related to the system's TKE, dissipation

rate, and the scalar Langevin isotropic drift coefficient, $\mathcal{G}$ as follows:

$$\boldsymbol{G}_{ij} = -\frac{\delta_{ij}}{T_L} := -\mathcal{G}\delta_{ij} \tag{18}$$

$T_L$ is also referred to as the "relaxation timescale" for turbulent mixing/spreading because it characterizes the time scale over which turbulent fluctuating velocity reverts to the background mean velocity. Lastly, in isotropic turbulence, the velocity variance $\sigma^2$ can be related to the turbulent kinetic energy $k$ as

$$\sigma^2 = \frac{2}{3}k, \tag{19}$$

which is equivalently stated as

$$\frac{2\sigma^2}{T_L} = C_0\epsilon. \tag{20}$$





Taken together, and imposing a scalar, isotropic $\boldsymbol{G} \equiv \mathcal{G}\delta_{ij}$, equations (15)-(18) result in the Langevin model we consider in this work, namely

$$\mathrm{d}U_i^*(t) = -\frac{1}{\rho}\frac{\partial P}{\partial x_i}^{0}\mathrm{d}t + \mathcal{G}^{-T_L^{-1}}\left[U_j^*(t) - \overline{U_j^*}\right]\mathrm{d}t + \sqrt{C_0\epsilon}\,\mathrm{d}W_i(t), \tag{21}$$

$$= \frac{\overline{U_i^*} - U_i^*}{T_L}\mathrm{d}t + \sqrt{\frac{2\sigma^2}{T_L}}\,\mathrm{d}W_i(t), \tag{22}$$

or in a more compact, vectorized notation,

$$\mathrm{d}\boldsymbol{U}_t^* = \underbrace{\frac{\overline{\boldsymbol{U}_t^*} - \boldsymbol{U}_t^*}{T_L}\mathrm{d}t}_{\text{drift term}} + \underbrace{\sqrt{\frac{2\sigma^2}{T_L}}\,\mathrm{d}\boldsymbol{W}_t}_{\text{Brownian motion term}}\,. \tag{23}$$

Here, the first bracketed term captures the memory effects of the relaxation timescale, over which the Lagrangian velocity
returns to the mean, and the second bracketed term is the Brownian motion contribution. The structure of (23) permits a broader range of behaviors than a traditional purely diffusive Gaussian plume model that represents turbulent mixing as a constant eddy diffusivity and is restricted to $\sqrt{t}$ growth (as discussed in Section 4.1). While the above equation is able to represent three-dimensional ($d = 3$) flows, we restrict the following analysis to the horizontal dimensions (in particular, the $x$-dimension) by vertically averaging boundary-layer quantities. Vertical averaging simplifies the particle model and focuses on horizontal
plume spreading, which is the most GCM-relevant given the vertical spreading of the plume throughout boundary-layer depth occurs on much shorter spatial and temporal scales. The vertical dimension reduction contains the implicit assumption that the boundary layer remains in an approximately well-mixed state, which is oftentimes the case in shallow, cloud-topped marine boundary layers.

## 2.3 Numerical Implementation of Langevin Particle Model

The particle model we consider is composed of a collection of $N_p$ particles that approximate the initial condition (IC) of the passive, conserved tracer (2) as a field composed of weighted kernel functions $k(\boldsymbol{x}, \boldsymbol{y})$. First, we note that, hereafter, we abandon the single arbitrary particle analysis applied in Section 2.2 and consider a full ensemble of particles, identified by subscript index $i \in [1, N_p]$. We formulate the model for the previously-described boundary layer flows in $d = 2$ spatial dimensions in the horizontal, or $xy$-plane. The IC may be formulated to be

$$\phi_0(\boldsymbol{x}) = \sum_{i=1}^{N_p}\int_{\mathbb{R}^2} m_i k(\boldsymbol{x} - \boldsymbol{z})\delta(\boldsymbol{z} - \boldsymbol{X}_i^*(0))\mathrm{d}\boldsymbol{z} \tag{24}$$

$$= \sum_{i=1}^{N_p} m_i k(\boldsymbol{x} - \boldsymbol{X}_i^*(0)). \tag{25}$$





We specify here that these kernels possess the standard properties that they are symmetric, translation-invariant, non-negative, and integrate to unity; i.e.,

$$k(\boldsymbol{x} - \boldsymbol{y}) \equiv k(\boldsymbol{y} - \boldsymbol{x}) := k(\boldsymbol{z}), \tag{26}$$

$$k(\boldsymbol{z}) \geq 0, \ \forall \boldsymbol{z} \in \mathbb{R}^2, \tag{27}$$

$$\int_{\mathbb{R}^2} k(\boldsymbol{z}) \mathrm{d}\boldsymbol{z} = 1. \tag{28}$$

The evolution of the tracer field is driven by the fluctuating Lagrangian velocities that follow (23), and the particle positions change according to

$$\frac{\mathrm{d}\boldsymbol{X}_i^*(t)}{\mathrm{d}t} = \boldsymbol{U}_i^*(t). \tag{29}$$

To solve the Langevin model (LM) governed by (23), we integrate the Langevin equation for velocity over a time step of length $\Delta t$ employing the Euler-Maruyama method

$$\boldsymbol{U}_i^*(t) = \boldsymbol{U}_i^*(t - \Delta t) + \frac{\overline{\boldsymbol{U}_i^*(t - \Delta t)} - \boldsymbol{U}_i^*(t - \Delta t)}{T_L} \Delta t + \sqrt{\frac{2\sigma^2 \Delta t}{T_L}} \, \boldsymbol{\xi}_i(t). \tag{30}$$

Above, $\sqrt{\Delta t} \, \boldsymbol{\xi}_i(t) = \Delta \boldsymbol{W}_i \approx d\boldsymbol{W}_i(t)$, and $\boldsymbol{\xi}_i$ is a 2-dimensional vector with independent and identically distributed entries drawn from a standard Normal distribution, i.e., $\boldsymbol{\xi}_i(t) \sim \mathcal{N}(\boldsymbol{0}, \boldsymbol{I})$. Finally, we use this updated Lagrangian velocity to update
particle positions, again using forward Euler, given by

$$\boldsymbol{X}_i^*(t + \Delta t) = \boldsymbol{X}_i^*(t) + \boldsymbol{U}_i^*(t)\Delta t. \tag{31}$$

## 3  Large-eddy simulation configuration used to inform particle model

The LES configuration in this study largely follows that of Chun et al. (2023), using version 6.10.9 of the System for Atmospheric Modeling (SAM; Khairoutdinov and Randall (2003)) with additional capabilities for representing the aerosol accumu-
lation mode developed at the University of Washington (UW-SAM; Berner et al. (2013)) to simulate an evolving ship plume with varying background aerosol conditions. The LES forcing profiles for the CONTROL simulation are derived from averaging shallow coastal stratocumulus boundary layers (400-800 m deep) in the northeast Pacific during July 2003 using ECMWF Interim Reanalysis (Zhang et al., 2012; Blossey et al., 2013), and the mean forcings are constant in time for the duration of the simulations. Surface fluxes of latent and sensible heat respond to local conditions according to Monin-Obukhov similarity the-
ory, and radiative transfer calculations were performed every 15 s (every 5 model time steps) using the Rapid Radiative Transfer Model for GCM applications (RRTMG; Mlawer et al. (1997)) with a diurnally varying zenith angle. The evolution of liquid hydrometeors (cloud and rain droplets) was handled by the two-moment Morrison microphysics parameterization (Morrison and Grabowski, 2008) and cloud water was diagnosed via saturation adjustment. The subgrid turbulence was represented using a 1.5-order TKE closure, which allowed for anisotropic diffusivity (Deardoff, 1980). Scalar advection was calculated using



the fifth-order Ultimate-Macho scheme (Yamaguchi et al., 2011) and momentum advection was calculated using second-order finite differencing. All simulations were run for 12 hours to fully develop turbulence overnight (spin-up period) and for an additional 39.5 hours post-ship injection, with the initial injection occurring just before sunrise. The injection time was chosen to maximize the impact of the aerosol-cloud interaction, as sunrise is associated with a diurnal peak in the precipitation rate (Wood, 2012) and injecting during the nighttime hours would result in unchanged shortwave radiatve forcing. The LES domain

is turned so that the mean wind blows from north to south and is translated along with the mean wind at $-10.5$ m s$^{-1}$, which is the strength of the mean wind at 315 m altitude). The ship traverses the domain at the east-west centerpoint (102.4 km) over a 50-minute period with a domain-relative speed of 10.5 m s$^{-1}$ and an aerosol injection rate of $10^{16}$ particles s$^{-1}$.

The vertical grid contains 144 levels with variable grid spacing from 15 m near the surface to 5 m in the cloud layer and in the vicinity of the inversion layer. Grid spacing in the free troposphere increases to near 70 m at the domain top (1.555

km). In this study, our focus is on the zonal (cross-wind) spreading of an injected ship plume and as such, high aspect ratio (bowling alley) domain geometries are necessary to maximize the amount of time in which it takes the injected plume to fill the entire domain and to limit east-west plume-edge interactions as a result of periodic lateral boundary conditions. Simulations in Chun et al. (2023) use a 96 km × 9.6 km bowling alley domain with a uniform grid spacing of 50 m. Initial test simulations for the CONTROL run were done on a 102.4 km × 25.6 km horizontal grid (50 m grid spacing), requiring nearly 450,000

core hours, which translates to roughly 9 days of computational time on 2,048 processors. After Gaussian curve fitting of the average boundary-layer aerosol concentration, the estimated $2\sigma$ ship plume width was approaching the domain size by hour 20 of the simulation, necessitating a wider $x$-dimension to prevent plume-edge overlap. Modeling a 204.8 km × 25.6 km domain at 50 m grid spacing is computationally cost prohibitive and demands a larger horizontal grid spacing (100 or 200 m). Tests of larger grid spacing on the 102.4 km × 25.6 km domain revealed grid spacing sensitivities to coarsening, most notably at 200 m

grid spacing (Figure 1a; Figure 2); although, characteristic mesoscale cell sizes as determined from LWP power spectra show minimal sensitivity to grid spacing (Figure 1b).

At $\Delta x = \Delta y = 100$ and 200 m, the LWP at the end of the spin-up period remains consistent with that for a 50 m grid spacing (Figure 2a), but coarser grid spacing results in larger boundary-layer aerosol concentrations, weaker surface precipitation, and slightly deeper boundary layers (Figure 2). In an attempt to reconcile the coarse grid simulations with the 50 m configuration

we apply "hyperdiffusion" to reduce entrainment rates and dampen grid-scale noise (Wyant et al., 2018) in the momentum equations using

$$\frac{\mathrm{d}\boldsymbol{U}}{\mathrm{d}t} = -\frac{(\Delta x \Delta y)^2}{\tau_{\mathrm{hyper}}} \nabla^4 \boldsymbol{U}, \tag{32}$$

where $\tau_{\mathrm{hyper}}$ is the diffusion timescale and has a value of 1200 s and 60 s for $\Delta x = \Delta y = 100$ and 200 m, respectively. For the 200 m simulation, strong hyperdiffusion effectively mutes entrainment resulting in excessive LWP and a shallow boundary-

layer depth, all while continuing to underestimate the surface precipitation in comparison to the 50 m run. The inability of the 200 m simulation to match 50 m surface precipitation despite larger LWP and in-line aerosol concentrations suggests that there may be grid sensitivity to precipitation rates independent of entrainment. Using 100 m horizontal grid spacing with





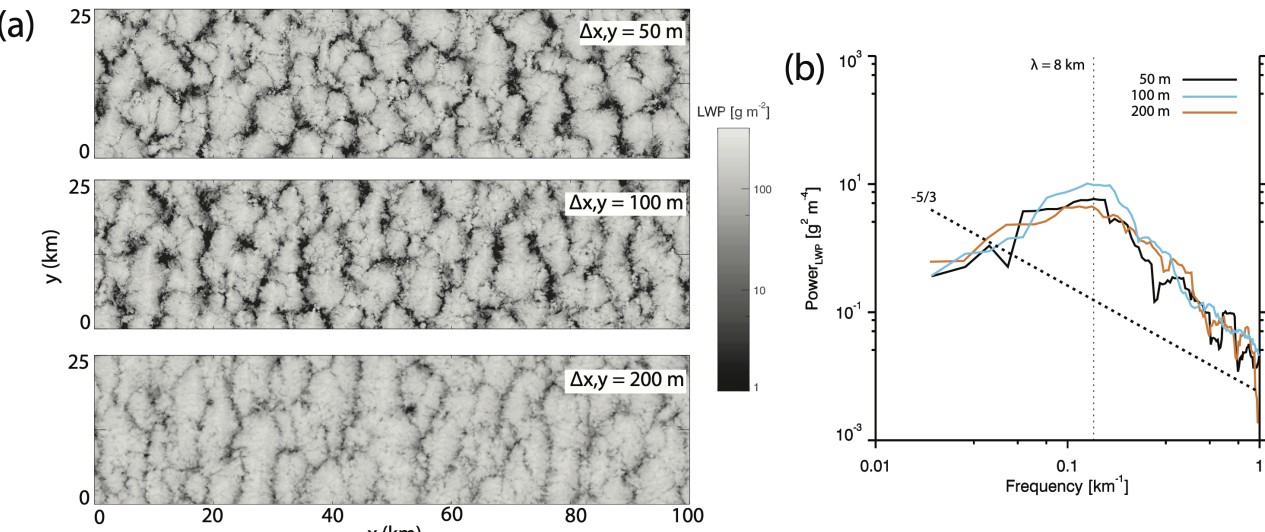

**Figure 1.** Left panel (a): LWP cross-sections at the end of the spin-up period (hr 12) in 100 km x 25 km simulations with 50 m horizontal grid spacing (top), 100 m grid spacing (middle), and 200 m grid spacing (bottom). Right panel (b): The smoothed LWP power spectra for the three different horizontal grid spacings. The sloped dashed line represents the -5/3 power law from satellite observations (Wood and Hartmann, 2006) and the vertical dashed line denotes the characteristic wavelength at the spectral peak ($\approx$ 8 km cell size).

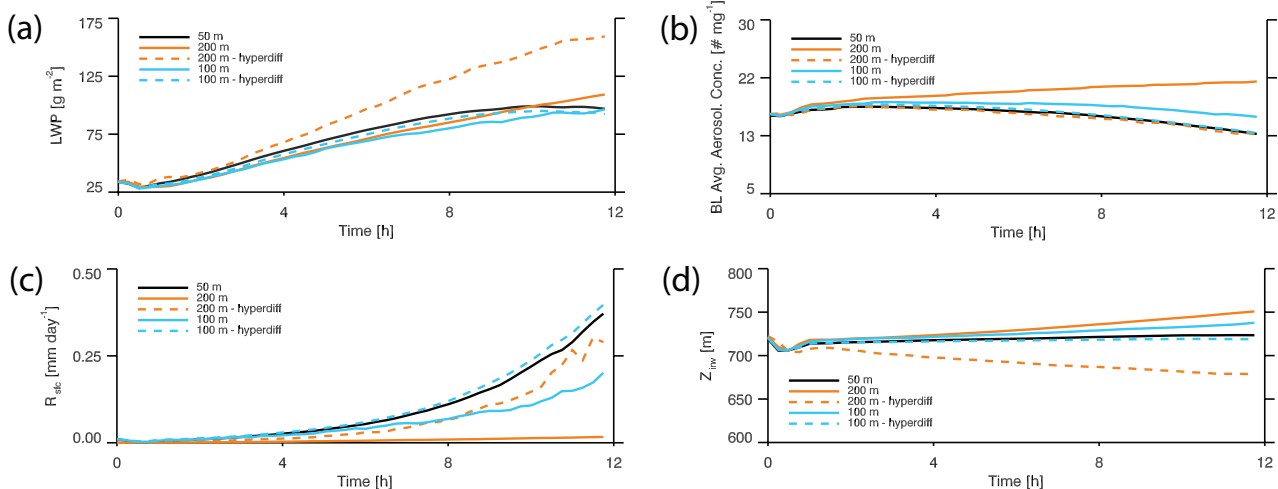

**Figure 2.** Spin-up period (first 12 hours of the simulation) time series for (a) LWP, (b) Boundary-layer-averaged aerosol concentration, (c) surface rain rate, and (d) inversion height. Dashed lines indicate that the simulation included hyperdiffusion.




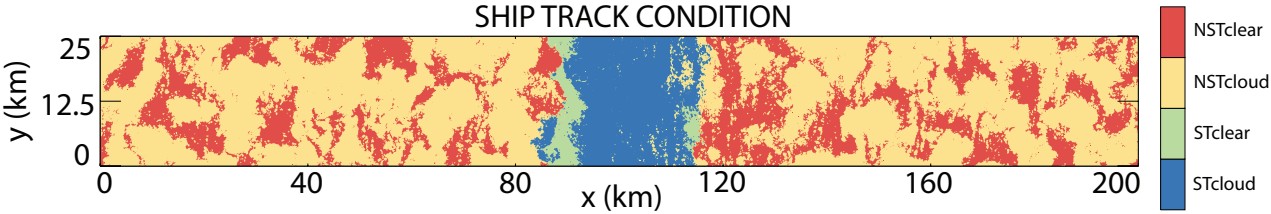

**Figure 3.** CONTROL run snapshot at 12 hours after ship injection (6 PM PDT) illustrating the different regions being conditionally sampled in the LES: STcloud - ship track and cloudy, STclear - ship track and not cloudy, NSTcloud - no ship track and cloudy, NSTclear - no ship track and not cloudy.

$\tau_{\text{hyper}} = 1200$ s results in boundary-layer depths, aerosol concentrations, rain rates, and LWP that are in agreement with the 50 m baseline simulation (Figure 2) and consequently, we make use of 100 m grid spacing with hyperdiffusion in order to use

sufficiently large domains (204.8 km × 25.6 km).

To explore the sensitivity of the particle model to input parameters from various regions within LES domain, the TKE, variances, and dissipation are conditionally sampled. The ship track (plume) is present if the grid-space-weighted aerosol concentration in the lowest 30 model grid levels is 3 times larger than the maximum column deviation. This stringent criteria ensures that non-ship columns are not incorrectly identified while potentially underestimating the fraction of the grid covered

by the ship plume. Figure 3 shows the four conditional statistic categories identified in the LES, with the main conditional average of interest in this study being the cloudy region with ship track present (STcloud or in-plume). Statistics were output every 15 minutes and 3-D files were output every 30 minutes.

Given the previously established spreading rate dependence on both wind shear (Berner et al., 2015) and precipitation (Prabhakaran et al., 2023), our sensitivity studies attempt to span a realizable range of zonal wind shear magnitudes and both

precipitating and non-precipitating cases to assess the ability of the particle model to represent a broad range of environmental conditions.

### 3.1  Shear sensitivity tests

Previous modeling studies of stratocumulus have generally focused on the impact of shear across the inversion layer that acts to deplete liquid water through enhanced entrainment and reduce overall TKE (Wang et al., 2008; McMichael et al., 2019;

Zapata et al., 2021), while in this study we wish to explore wind shear "within" the boundary layer. The span of zonal shear (cross-plume wind) magnitudes in this study was achieved by altering the geostrophic wind profile to land at a post-spin-up shear that does not dramatically alter vertical shear near cloud top and restricts the bulk of the wind shear to the subcloud layer. In addition to adjusting the forcing wind profile, the weak shear case (WEAK) neglects the apparent Coriolis force. To better constrain zonal shear magnitudes in our simulations, we analyze 2,208 Lagrangian trajectories from 6 different source regions

in the northeast Pacific during June-August, 2018-2021 using ECMWF Reanalysis v5 (ERA5) (Eastman and Wood, 2016; Mohrmann et al., 2019; Erfani et al., 2022). Restricting the analysis to grid levels in ERA5 may result in substantial errors and





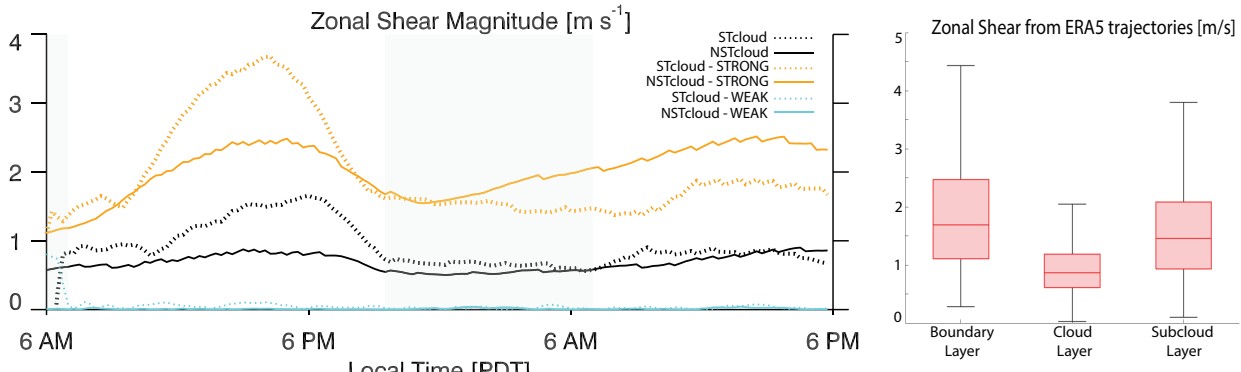

**Figure 4.** Left plot: Time series of boundary-layer zonal shear magnitude in the cloudy, ship track region (STcloud; dashed lines) and the cloudy, non-ship environment (NSTclear; solid lines). 6 AM corresponds to the initial injection time. Right plot: Maximum zonal shear magnitudes estimated from ERA5 Lagrangian trajectories in the northeast Pacific.

we instead linearly interpolate the zonal wind to the estimated boundary-layer and cloud-base heights. The surface wind speed is approximated by the 10 m wind speed. If the cloud base is found to be below the estimated boundary-layer height and the cloud depth exceeds 50 m we compute the boundary-layer, cloud-layer, and subcloud-layer zonal shear magnitudes. Median maximum boundary-layer zonal shear magnitudes in the ERA5 trajectories are $\approx 1.7 \text{m s}^{-1}$, with median subcloud layer shear being almost double that of the cloud layer (Figure 4).

The CONTROL case maximum zonal shear magnitude in the NSTcloud region is 0.9 m s$^{-1}$ which is near the 25th percentile of shear magnitudes from ERA5 trajectories, with shear magnitudes of up to 1.5 m s$^{-1}$ in STcloud during the first evening period when weak decoupling occurs (Figure 4). For the strong shear case (STRONG), maximum zonal shear magnitudes in NSTcloud are 2.5 m s$^{-1}$ (75th percentile) with STcloud shear magnitudes exceeding 3.5 m s$^{-1}$ (>90th percentile). During the overnight period, the STRONG case zonal shear in STcloud becomes weaker than the non-plume environment and remains weaker for the duration of the run (Figure 4). The WEAK case maintains less than 0.1 m s$^{-1}$ of zonal shear, representing an extreme case of low wind shear not seen in the ERA5 trajectories (Figure 4).

### 3.1.1 Macroscale evolution of shear simulations

The varying shear magnitudes result in broadly similar $x$-$y$ LWP evolution with no conspicuous signal of a ship perturbation one hour after the initial injection (Figure 5). All shear simulations exhibit cloud clearing near the ship-plume edge (with the plume-edge being easily identifiable in Figure 6) during the first evening period (hour 13) with the clear region recovering by the following morning (hour 25) (Figure 5). This cloud-clearing feature arises from a buoyancy anomaly in the ship-track region that induces a mesoscale circulation and results in subsidence near the plume edge that creates an inhospitable environment for cloud development (Chun et al., 2023; Wang et al., 2011; Prabhakaran et al., 2023). The CONTROL and STRONG cases initially contain homogeneous, closed-cell convection which transitions to more broken cloud conditions in the







**Figure 5.** Liquid water path (LWP) evolution of the four large-domain LESs from a bird's-eye view ($x$-$y$ plane) beginning one hour after ship injection (7 AM) and at 12 hour intervals thereafter. Zonal shear vector in CONTROL and STRONG is from right to left.

non-ship region, with the ship track region maintaining near-overcast conditions for the first 25 hours (Figure 5). The overcast ship track region begins to break up during the second evening period, with the STRONG and WEAK cases experiencing more severe fragmentation than the CONTROL (Figure 5). Non-ship-region mesoscale LWP structure is notably different in the WEAK case, with narrow bands of cloud from the first evening onward, indicative of open-cell convection (Figure 5). Precipitation suppression brought about by the increase of CCN is evident in all shear simulations (Figure 7). At one hour after injection, the WEAK case has the most intense precipitating cells (Figure 7), with a decrease in precipitation intensity later in the day in all cases, which is consistent with the typical diurnal cycle of stratocumulus precipitation (Wood, 2012). Local precipitation enhancement occurs on the down-shear side of the plume edge in the CONTROL and STRONG case (Figure 7). The background aerosol concentration is modified by the precipitation rate through scavenging, with lower environmental aerosol concentrations during the early morning hours, coinciding with the strongest precipitation (Figure 6, 7). Background aerosol is able to recover during the second evening as entrainment and surface aerosol sources are larger than the aerosol losses from precipitation (Figure 6).



## CONTROL

## STRONG SHEAR

## WEAK SHEAR

## POLLUTED

**Figure 6.** Average dry aerosol number concentration of the bottom 30 grid levels for the four large-domain LESs from a bird's-eye view ($x$-$y$ plane) beginning one hour after ship injection (7 AM) and at 12 hour intervals thereafter.

The CONTROL and STRONG cases have comparable domain-averaged LWP, inversion heights, entrainment rates, surface precipitation rates, and surface fluxes during the first 15 hours of the simulations, with the strongest divergence during the first overnight period as the STRONG case experiences less domain-averaged entrainment and an attendant lower average inversion height (Figure 8). Surface precipitation rates in the WEAK case are more than twice as large as the CONTROL which depletes LWP, reduces the entrainment rate, and results in boundary layer that is $\approx 15\%$ shallower than the CONTROL. Surface sensible heat fluxes remain similar between the shear cases, although surface latent heat fluxes in the WEAK case are $\approx 20\%$ smaller than the CONTROL.

### 3.2 Background aerosol sensitivity test

For the CONTROL case, the initial boundary-layer aerosol concentration is 20 # mg$^{-1}$ and the free-tropospheric aerosol concentration is 50 # mg$^{-1}$. For the POLLUTED case, the initial boundary-layer aerosol concentration is 130 # mg$^{-1}$ and the free-tropospheric aerosol concentration is 100 # mg$^{-1}$. The injection rate in the POLLUTED case is increased to $3.25 \times 10^{16}$







**Figure 7.** Surface precipitation rate evolution for the four large-domain LESs from a bird's-eye view ($x$-$y$ plane) beginning one hour after ship injection (7 AM) and at 12 hour intervals thereafter. Zonal shear vector in CONTROL and STRONG is from right to left.

s$^{-1}$ to generate an aerosol perturbation that is roughly the same size as the CONTROL on a percentage basis (Chun et al., 2023). The POLLUTED case large-scale subsidence is increased by 50% in comparison to the CONTROL as a means to attain a similar boundary-layer depth in the presence of much larger LWP and entrainment rates (Figure 8).

### 3.2.1 Macroscale evolution of polluted simulation

The POLLUTED case has a markedly different mesoscale structure than the three shear cases, with cloud fraction near unity

and convective cells that remain closed for the duration of the simulation, along with no discernible ship track region in the LWP field (Figure 5). While background aerosol concentrations in the shear cases decrease during the overnight period as a result of precipitation scavenging, the decrease in average aerosol concentration in the POLLUTED case is a result of the free troposphere being cleaner than the boundary-layer and the entrainment acting as a sink instead of a source (Figure 6). The increased aerosol concentration in the POLLUTED case is sufficient to shut down precipitation production for the duration of



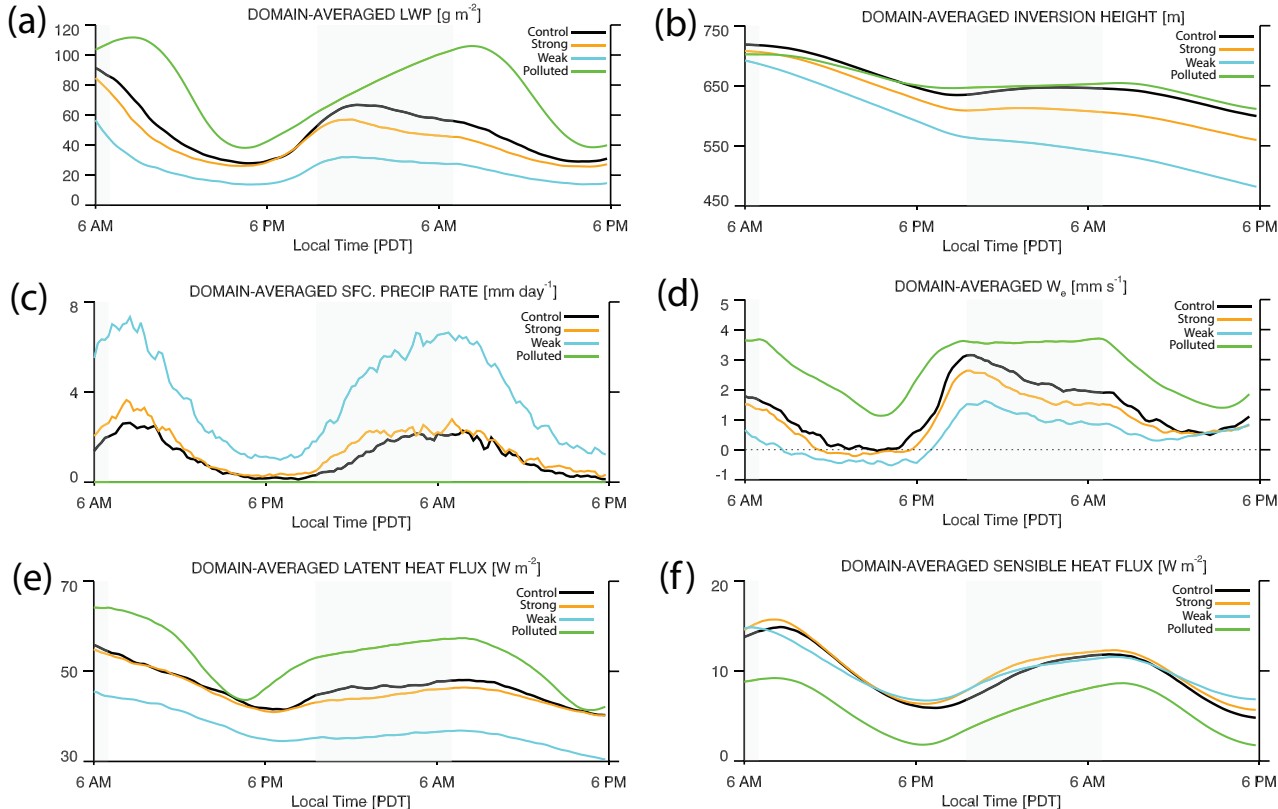

**Figure 8.** Domain-averaged time series of (a) Liquid water path, (b) inversion height, (c) surface precipitation rate, (d) entrainment rate, (e) surface latent heat flux, and (f) surface sensible heat flux. 6 AM corresponds to the initial injection time. Gray shaded region is the time period with no shortwave radiation present.

the simulation (Figure 7) and provides an opportunity to explore the ability of the particle model to represent precipitation- or aerosol-concentration-induced responses.

### 3.3   Langevin particle model input parameters

#### 3.3.1   Domain-averaged input parameters

Zonal variances fluctuate modestly diurnally, with a $\approx 20\%$ reduction in zonal variance from the overnight period into the

evening period as the buoyancy production of TKE driven by radiative cooling near cloud-top is stunted by solar absorption (Figure 8a). The meridional variance remains nearly constant for the entire simulation. The diurnal signal is most evident in the vertical velocity variance, with the variance decreasing by nearly 50% by 4 PM of the first day (Figure 8c). Vertical velocity variance ramps up after 6 PM as solar insolation decreases (Figure 8c). The dissipation rate generally mirrors the TKE, with a minimum in dissipation rate in the evening with greater dissipation overnight associated with invigorated turbulence (Figure





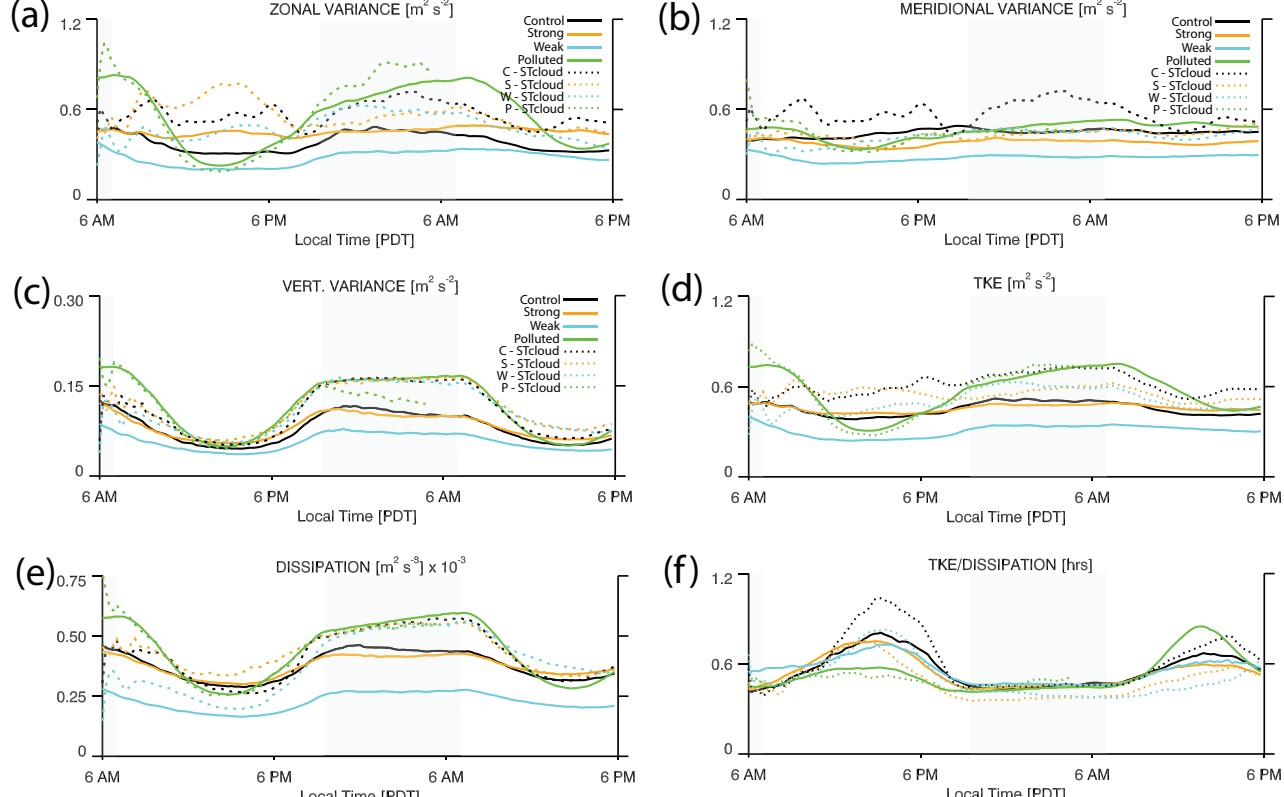

**Figure 9.** Turbulence-related input parameter time series relevant to the Langevin particle model: (a) zonal (east-west) velocity variance, (b) meridional (north-south) velocity variance, (c) vertical velocity variance, (d) the resolved TKE, (e) the resolved dissipation rate ($\epsilon$), and (f) the relaxation timescale ($T_L$) without the $C_0$ correction. All quantities are averaged over the entire boundary-layer depth. Solid lines correspond to domain-averaged time series and dashed lines correspond to in-plume (STcloud) averaged time series.

8d,e); however, the relationship between TKE and dissipation can be complex and vary in space and time, depending on local flow geometry, turbulent length scales and stratification. The TKE divided by dissipation, $k/\epsilon$, yields a timescale, which is related to the relaxation of a particle's velocity to the mean velocity, and longer relaxation timescales ($T_L$) can be interpreted as a longer leash on any individual particle, allowing the particle to deviate farther from the mean wind profile before being tugged back strongly. The relaxation timescale for the CONTROL run peaks around 3 PM and is smallest and nearly constant

during the night (Figure 8f).

The STRONG case domain-averaged zonal and meridional velocity variance shows little evidence of a diurnal cycle with nearly constant variance for the entire period (Figure 8a,b). Vertical velocity variance does have a diurnal cycle and is almost indistinguishable from that of the CONTROL simulation (Figure 8c). The overall TKE remains similar between the two cases, despite the STRONG case having a slightly dampened diurnal cycle (Figure 8d). From 3 pm until sunset the first day, the

nearly equal TKE between the CONTROL and STRONG case coupled with a STRONG case dissipation rate that is marginally



larger gives rise to a smaller relaxation timescale in the STRONG case (Figure 8d,e,f). The WEAK case has an analogous diurnal cycle in zonal variance to the CONTROL, but the magnitude of all components of the variance is lower (Figure 8a,b,c). The evening restrengthening of vertical velocity variance is less pronounced in the WEAK case but the relaxation timescale is in-line with the CONTROL and STRONG case.

The diurnal cycle of zonal and vertical velocity variance is much larger in the POLLUTED case (Figures 8a,c), with zonal variances that plummet during the afternoon to values seen in the WEAK case and a rebound in zonal variance overnight that is nearly double that of the CONTROL by the early morning hours of day 2 (Figure 8a). The POLLUTED case relaxation timescale remains lower than all of the shear cases during the first day, with comparable relaxation timescales overnight (Figure 8f). During the second day, the POLLUTED case relaxation timescale is considerably larger than the three other cases, 395  suggesting a different relationship between TKE and dissipation from day 1 to day 2 (Figure 8f).

### 3.3.2 In-plume-averaged input parameters

Now focusing on the conditionally-sampled ship track region (STcloud), the CONTROL run experiences enhanced TKE in comparison to the domain average (Figure 8d). The diurnal cycle of zonal velocity variance is not as apparent in the ship plume, while the diurnal cycle of vertical velocity variance is magnified with large increases in vertical variance overnight 400  (Figure 8a,c). This is consistent with the findings of Chun et al. (2023), where plume turbulence intensification through the suppression of drizzle dominated over other potential factors such as increased entrainment efficiency in the ship track region. Although the TKE in the ship track region is larger than the domain-averaged TKE the dissipation is smaller during the evening of day 1 (Figure 8d,e), suggesting that the energy cascade at small scales is fundamentally different and the plume turbulence may be dominated by larger coherent structures that dissipate energy less efficiently, such as a mesoscale circulation driven by 405  precipitation suppression. Additionally, organized horizontal TKE transport in the ship region region may be causing the TKE to be dissipated in the non-ship region. In comparison to the non-ship track region, the ship track region experiences larger negative buoyancy fluxes near cloud base and stronger daytime decoupling, leading to longer ship-track relaxation timescales as a result of less efficient dissipation. The relaxation timescale in the plume is $\approx 25\%$ longer than the domain average during the first daytime period but converges to the domain-averaged relaxation timescale during the overnight period, pointing to a return 410  to more isotropic, homogeneous turbulence. The presence of decoupling challenges the well-mixed assumption; however, LES spreading rates in the subcloud and cloud layers differ negligibly from the boundary-layer averaged spreading rate.

    Similar to the CONTROL case, the WEAK and STRONG cases both show stronger turbulence in STcloud in comparison to the domain averages (Figure 9). The STRONG case has a peak in zonal velocity variance around 4 PM while the WEAK case zonal variance peaks during the overnight hours (Figure 9a). In-plume relaxation timescales in the STRONG and WEAK cases 415  are lower than their respective domain averages after 6 PM of day 1, indicating a more efficient dissipation of energy from then on.

    As a consequence of being non-precipitating, the POLLUTED case does not exhibit large differences between in-plume and domain-averaged turbulent statistics over the first 15 hours (Wang et al., 2011; Chun et al., 2023; Prabhakaran et al., 2023). In-





plume data points after hour 15 are questionable given that conditionally-sampled plume fraction underwent a rapid decrease

as the aerosol concentration criteria established before ship injection became too restrictive for the POLLUTED case.

### 3.4 Plume width calculation for particle model comparison

The particle model lends itself to easily identifying the $1\sigma$ plume width simply by computing the standard deviation of the particle-position PDF and multiplying by two to obtain the full width. The LES plume width is complicated by the presence of the background aerosol variability and we employ a Gaussian curve fitting procedure on the boundary-layer-average aerosol

concentration to define the $1\sigma$ plume width. We use the following curve fitting equation

$$f(x) = A \exp\left[-\frac{(x-B)^2}{2C^2}\right] + D, \tag{33}$$

where $A$ represents the the amplitude of the aerosol perturbation ($\#\,\mathrm{mg}^{-1}$) in the ship plume, $x$ is the position along the $x$-axis (km), $B$ is the location of the ship plume center (km), $C$ is the standard deviation (km), and $D$ is the background aerosol concentration ($\#\,\mathrm{mg}^{-1}$). The algorithm must be provided with initial guesses of $A, B, C, D$, which were $[130.0, 102.4, 8.0, 20.0]$

for the CONTROL and shear cases and $[350.0, 102.4, 8.0, 130.0]$ for the POLLUTED case. Gaussian curve fitting along the $x$-dimension was performed at each $y$-location and then averaged along the entire $y$-dimension to create a single $1\sigma$ plume width estimate, which is then multiplied by two to get an actual plume width.

## 4 Results

### 4.1 Large-eddy simulation plume width results

For an initial delta function injection and purely diffusive plume spreading, we define the time-varying, expected width of the plume as

$$\mathcal{W}(t) = 2\sqrt{2\mathcal{D}t}, \tag{34}$$

where $\mathcal{D}$ is the coefficient for eddy diffusivity and $t$ is time.

When analyzing the eddy diffusivity output from the 1.5-order TKE closure in the LES, maximum subgrid diffusivity values

are near $0.75\,\mathrm{m^2\,s^{-1}}$, which results in a plume width of 500 m at hour 10 after injection while the CONTROL $1\sigma$ plume width at hour 10 is 24 km (Figure 10). It is then immediately obvious that the mixing of the aerosol is not only being done by the smallest scales. As an alternative calculation, $\mathcal{D}$ can be estimated as the product of a characteristic zonal eddy velocity ($u_c$) and a characteristic length scale ($l$) on which the mixing occurs ($\mathcal{D} = u_c l$). In the CONTROL run, the characteristic zonal eddy velocity is $\approx 0.3\,\mathrm{m\,s^{-1}}$ and the characteristic mixing length scale may potentially be driven by mesoscale eddies which are

initially $\approx 8$ km wide. The resulting constant Gaussian diffusion equation estimates a width of 26.3 km at hour 10, which is much closer to the LES plume width (Figure 10). While assuming the larger scales are doing a majority of the mixing related to the ship track spreading does improve the simple Gaussian diffusion model performance, it fails to capture the linear to




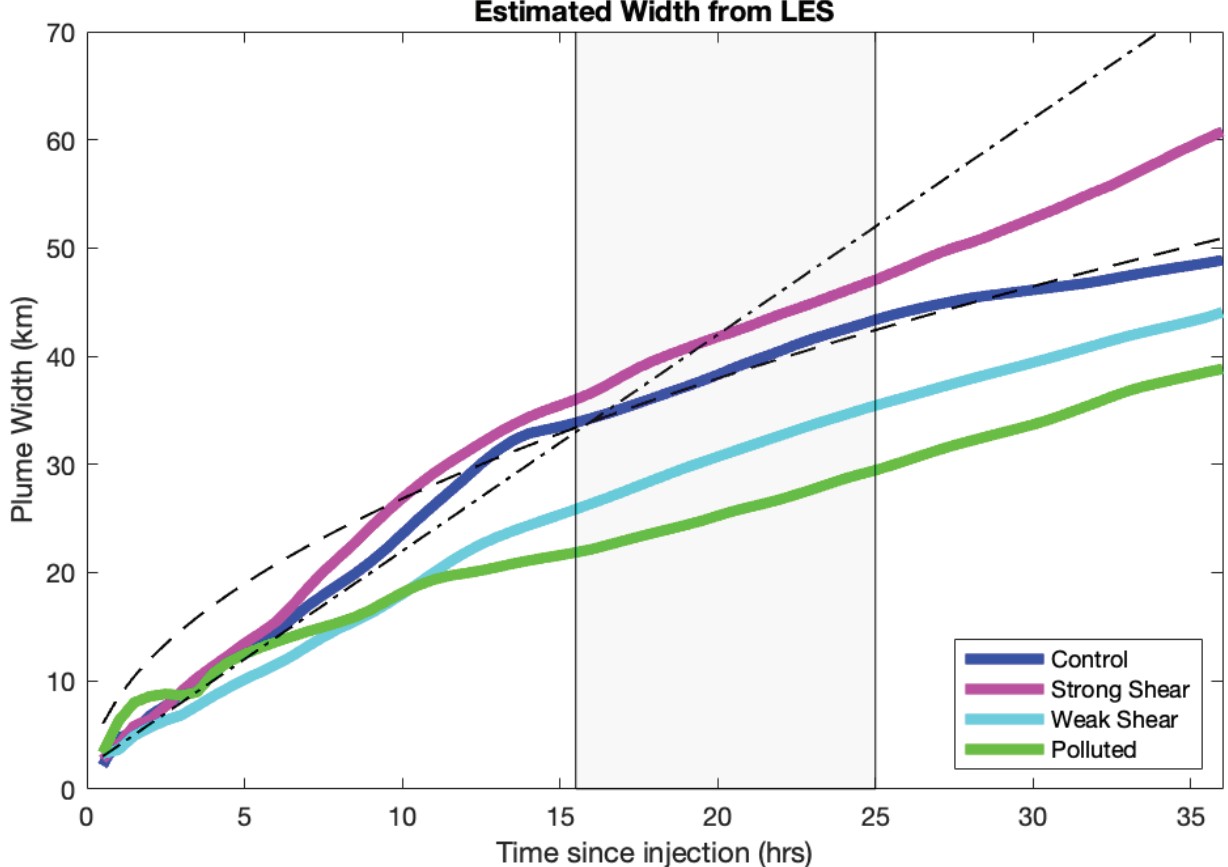

**Figure 10.** The one standard deviation (multiplied by 2 to get plume width) width estimated from the Gaussian curve fitting procedure applied to the LES aerosol concentrations at 30-minute intervals. Hour 0 corresponds to 6 AM. The dot-dashed line represents a constant plume growth rate of 2 km/hr (Heffter, 1965; Durkee et al., 2000). The dashed line represents a Gaussian diffusion curve (34) where $u_c = 0.3$ m s$^{-1}$ and $l = 8$ km. The shaded region is the nighttime period.

superlinear growth in the first 15 hours after injection (Figure 10). Given the inaccuracies of the Gaussian diffusion assumption in the first 15 hours along with uncertainties regarding the characteristic velocity and time-dependent characteristic mixing
length scales, an approach tied to the turbulence properties in the LES is desired and motivates the model developed in Section 2 .

    The CONTROL run plume grows at 2-3 km hr$^{-1}$ during the first 13 hours before decreasing around sunset and remaining near 1 km hr$^{-1}$ during the night. Further reductions in plume growth ($< 1$ km hr$^{-1}$) during day 2 lead to a final plume width of 48.8 km (Figure 10). The STRONG case plume growth remains similar to the CONTROL for the first 5 hours after injection
before a burst of 3 km hr$^{-1}$ growth (Figure 10) and by hour 10 the STRONG case plume is $> 3$ km wider than the CONTROL. Similarly to the CONTROL, the STRONG case plume growth rate lessens during the first evening and remains nearly constant



at 1 km hr$^{-1}$ throughout the night and into day 2 period, resulting in a final plume width of 60.8 km (Figure 10). WEAK case growth rates are initially slower than the CONTROL case with a plume width of 18 km at hour 10 (6 km narrower than CONTROL), but overnight spreading rates are consistent with the STRONG and CONTROL cases (Figure 10). Day 2 WEAK case plume spreading persists at $\approx$ 1 km hr$^{-1}$ and the final plume width is 44 km. The POLLUTED case experiences rapid growth in the first few hours after injection but average spreading rates between hours 5-15 are slower in comparison to the three other cases (Figure 10) and consistent with previous modeling studies indicating slower growth in non-precipitating boundary layers (Prabhakaran et al., 2023). Again, overnight and Day 2 spreading rates for the POLLUTED case are $\approx$ 1 km hr$^{-1}$, suggesting overnight horizontal spreading rates may be independent of zonal variance intensity. In the sensitivity cases examined here, the difference in LES plume width at hour 15 between the STRONG and POLLUTED cases is 14 km, emphasizing the importance of accounting for different background conditions when estimating plume growth rates.

## 4.2 Particle model results for sheared cases

The particle model requires initial conditions for both particle position and particle velocity, with the standard deviation of particle position being initialized as half of the first available plume width in the LES and the standard deviation of particle velocity being set to 0. Additionally, the particle model contains a free parameter, $C_0$, that is found by spanning a range of $C_0$ values using 20,000 particle simulations with 2-minute time steps and determining the value associated with the minimum cumulative least squares error compared to the LES. Since we have two different input parameter categories (domain average and in-plume) we find two distinct $C_0$ values for the CONTROL, which are $C_0 = 0.37$ for domain-averaged statistics and $C_0 = 0.69$ for the in-plume statistics. These constants are then applied to the sensitivity cases in hopes of being physically representative of broad-ranging turbulence behavior. By running numerous ensemble members with a low particle number, the number of total particles modeled can be reduced to near 1,000 (50 ensemble members, 20 particles each) with minimal deterioration in model performance and consistency. The results shown here use 50 ensemble members of 100 particles each, with each separate realization of the ensemble mean being stable and in agreement with simulations with 20,000 particles (or greater).

The domain-averaged input parameters are able to largely capture the plume spreading throughout the entire simulation, with errors at any point not exceeding 2.5 km (Figure 11). While the particle model broadly captures the spread, subtleties such as the downward inflection in the spreading rate near hour 13 are only captured when the model is forced with turbulence data from within the ship plume (Figure 11b). Neither the domain averages or in-plume averages capture the day 2 slowing of plume growth shortly after sunrise (near hour 25) (Figure 11). When the particle model is applied to the two shear sensitivity cases the performance becomes case-dependent. Both the domain-averaged and in-plume particle model forcings are able to time the deceleration in spreading rate adeptly, though plume widths in the STRONG case are underestimated substantially (Figure 12).

It is desirable for the particle model to capture the clear divergence in LES spreading rates between the STRONG case and CONTROL between hours 7$-$10 after injection. Zonal variance in the STRONG case is larger during this time frame and on its own would result in faster spreading; however, the STRONG case relaxation timescale is equal to or less than that of



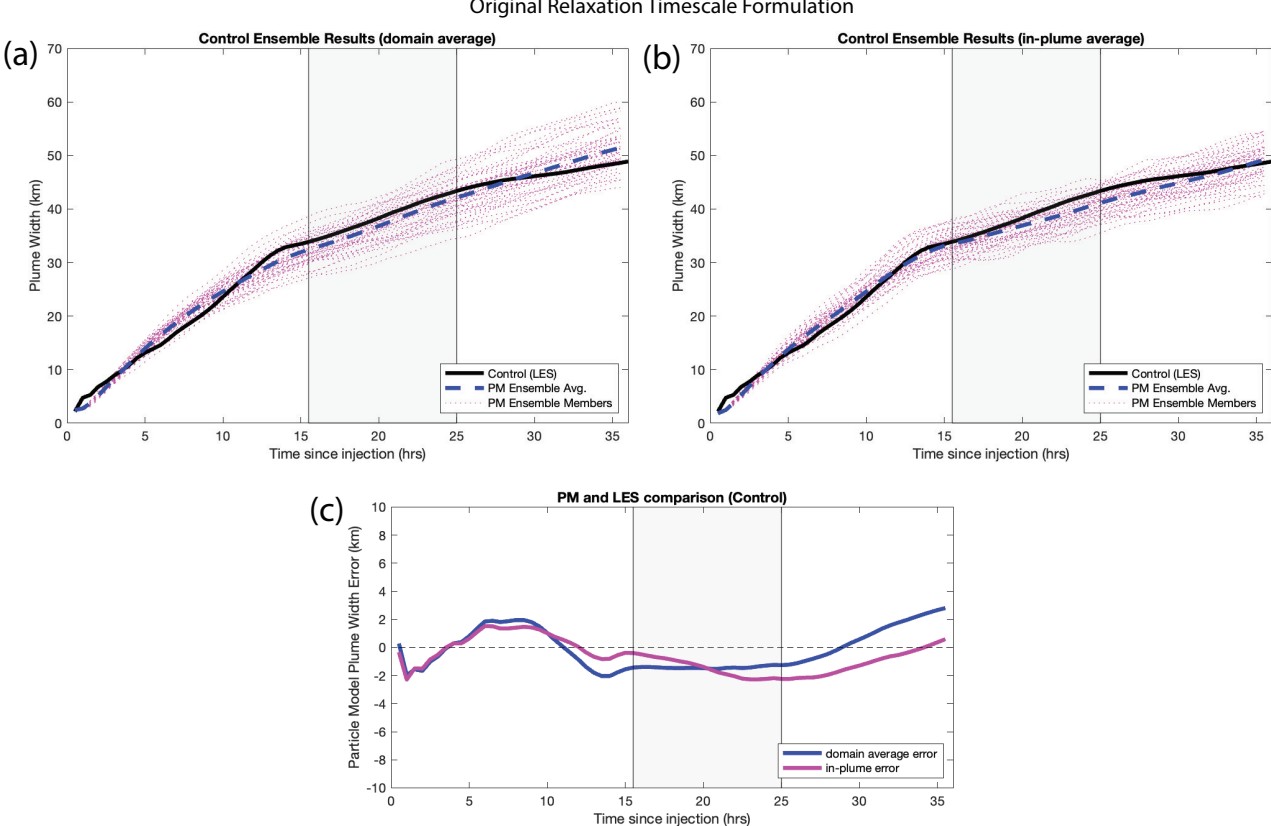

**Figure 11.** Ensemble particle model results for the CONTROL run using the original relaxation timescale formulation ($T_L$) laid out in Eq. 18. (a) Particle model results using the domain-averaged input parameters, (b) particle model results using the in-plume (STcloud) averages, and (c) the error between the particle model and the LES plume width for domain-averaged and in-plume input parameters. Individual dashed lines represent each ensemble member.

the CONTROL, leading to particle model spreading rates that are nearly identical. The original formulation of the relaxation timescale assumes isotropic turbulence, but we are instead interested in the variance along the dimension of the spread, which in this case is the zonal variance. We introduce a modified relaxation timescale calculation ($T_m$) that only considers the spread-dimension variance but also maintains an assumption of isotropic dissipation

495
$$T_m = \frac{\frac{1}{2}u'^2}{\frac{3}{4}C_m\epsilon},$$
(35)

where $u'^2$ is the zonal variance and $C_m$ is a new constant that must be found through another round of CONTROL run optimization. The $C_m$ constant is 0.15 for the domain-averaged forcing and 0.29 for the in-plume forcing. The new $C_m$ values account for the magnitude of $T_m$ being smaller in comparison to $T_L$, but the diurnal cycle of $T_m$ is more amplified in the STRONG case (Figure 13).



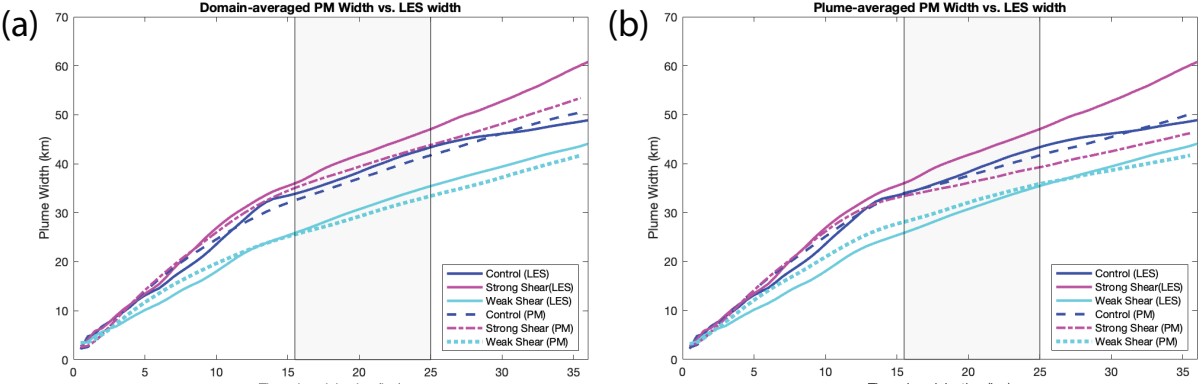

**Figure 12.** Ensemble particle model results (only ensemble averages shown) from the particle model are dashed lines and solid lines are LES plume widths. (a) Particle model forced with domain-averaged statistics and (b) particle model forced with in-plume (STcloud) statistics. $C_0$ is 0.37 and 0.69 for the domain-averaged and in-plume runs, respectively. The original relaxation timescale formulation ($T_L$) was used.

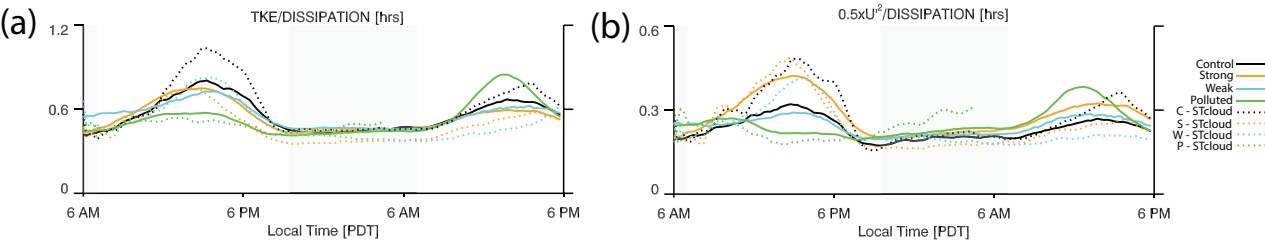

**Figure 13.** (a) The original relaxation timescale formulation ($T_L$) and (b) the modified version of the relaxation timescale ($T_m$), which focuses only on the zonal variance.

The use of $T_m$ resolves the relatively slow spread in the STRONG case during the $7-10$ hour period, with final plume widths in line with the LES when using domain-averaged forcings. Changing $T_m$ does not appreciably change the CONTROL or WEAK case results using domain average or in-plume statistics (Figure 14). Using $T_m$ with in-plume forcing results in good performance through hour 15 followed by poor performance during the nighttime and day 2 periods (Figure 14b). By applying $T_m$ and $C_m = 0.15$, the CONTROL, STRONG, and WEAK case particle model results all agree well with LES plume widths, with changes in diurnal-cycle-related spreading rates being captured using only domain-averaged turbulent statistics.

## 4.3 Particle model results for polluted case

The POLLUTED case forced with domain-averaged statistics, using both $T_L$ and $T_m$ performs poorly in comparison to LES plume width with growth that is much too extreme in the first 10 hours of the particle simulation (Figure 15), signifying that $C_0$ and $C_m$ are not translating as "universal" constants to the non-precipitating, POLLUTED case. All precipitating cases (CONTROL, STRONG, WEAK) develop mesoscale circulations shortly after ship injection that intensify until sunset, when





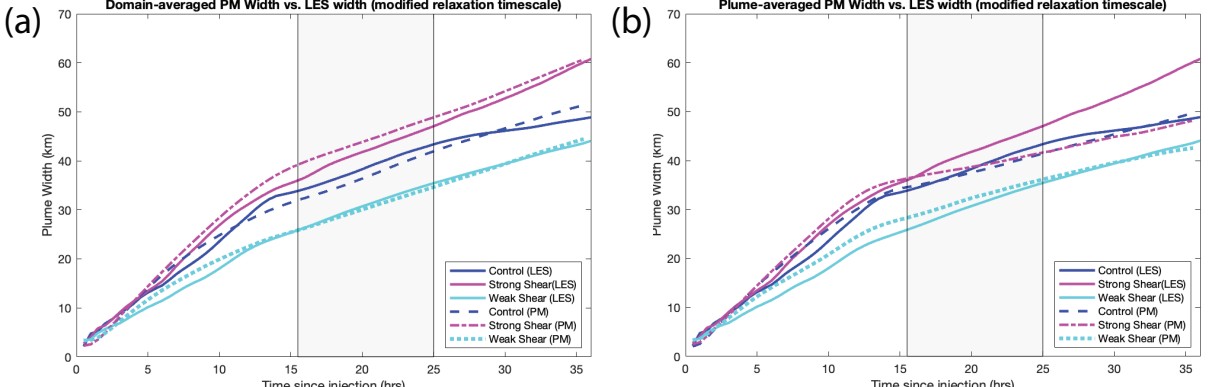

**Figure 14.** Ensemble particle model results (ensemble averages only shown) from the particle model using the modified relaxation timescale ($T_m$). Particle model results are dashed lines and solid lines are LES plume widths. (a) Particle model forced with domain-averaged statistics and (b) particle model forced with in-plume (STcloud) statistics. $C_m$ is 0.15 and 0.29 for the domain-averaged and in-plume runs, respectively.

stronger boundary-layer turbulence interferes with the circulation (Figure 16). These mesoscale circulations are believed to arise from the suppression of precipitation and the associated buoyancy anomaly in the ship track region (Prabhakaran et al., 2023; Chun et al., 2023; Wang et al., 2011), but what mechanisms sustain, intensify, or destroy them is currently unknown. Even in the absence of a complete understanding of mesoscale circulation dynamics, it remains a critical deviation from homogeneous, isotropic turbulence for which the particle model was developed. In this sense, $C_0$ and $C_m$ are specifically optimized to represent cases with mesoscale circulations present. Increased spreading rates are achieved through a reduction of isotropic dissipation in (35) through multiplication with ($C_0, C_m$), allowing for more rapid spread than would otherwise be possible given the domain-averaged forcings. Accounting for anisotropy with the Langevin model is commonly done by adding a correction term dependent on spatial derivatives of the variance (Legg and Raupach, 1982; Dehbi, 2008); however, such a correction is not desirable given that the purpose of this simplified model is to be used as a subgrid parameterization wherein variance gradients are not accessible. Therefore, we optimize $C_0$ for the POLLUTED case separately and propose the use of two constants for domain-averaged statistics using $T_m$: one for precipitating cases where mesoscale circulations develop ($C_p = 0.15$) and one for non-precipitating cases where turbulence remains nearly isotropic ($C_{np} = 0.50$). By correcting the POLLUTED case to not implicitly represent a mesoscale circulation the particle model is aligned with the LES width.

## 5 Conclusions

Estimating subgrid plume fraction in a climate model grid box necessitates a method of approximating the rate at which the plumes grow. Neglecting such growth, or assuming constant or Gaussian-diffusion-based growth, may cause non-linear errors that lead to unreliable or unrealistic responses to MCB. We employ a Lagrangian particle model, driven by large-domain LES





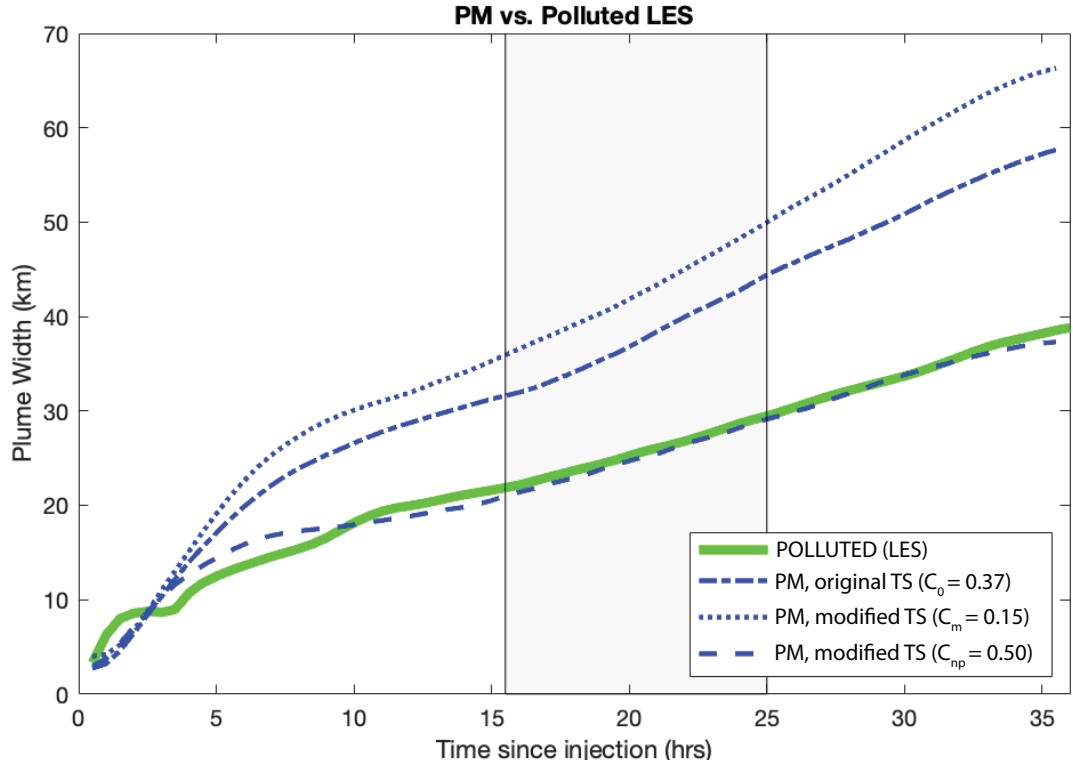

**Figure 15.** Ensemble average particle model result for the POLLUTED case. Particle model results are dashed lines and solid line is the LES plume widths. PM ($C_0 = 0.37$) is using the original relaxation timescale formulation $T_L$ and $C_0$. PM, modified TS ($C_m = 0.15$) uses $T_m$ and the CONTROL optimized $C_m$. PM, modified TS ($C_{np} = 0.50$) uses $T_m$ and a new optimized constant $C_{np}$ that applies to non-precipitating cases.

output of ship tracks, and assess the reduced-order model's ability to represent plume spreading in environments with different
shear magnitudes and environments with or without precipitation.

Using only 5000 particles (50 ensemble members with 100 particles) in the naturally-parallel stochastic particle model, both the domain average and conditionally-sampled plume TKE, variances, and dissipation rates result in good agreement with the LES CONTROL plume width, with the in-plume statistics more accurately timing the slowdown of spreading rate as the mesoscale circulation weakened near sunset on day 1. In-plume statistics perform better than domain-averaged quantities
during the first 15 hours after injection, but domain-averaged statistics perform best during the overnight and day 2 periods. Extending the CONTROL-optimized free parameter $C_0$ to the STRONG case resulted in poor performance as the original relaxation timescale formulation $T_L$ assumes that each component of the variance is equally contributing to the zonal spread. Instead, we apply a modified relaxation timescale $T_m$ which focuses on the variance only in the direction of the spread (zonal direction in this study). This modified formulation results in appropriately larger spreading rates in the STRONG case and
minimally impacts the CONTROL and WEAK case spreading rates. The particle model formulation that performs best in





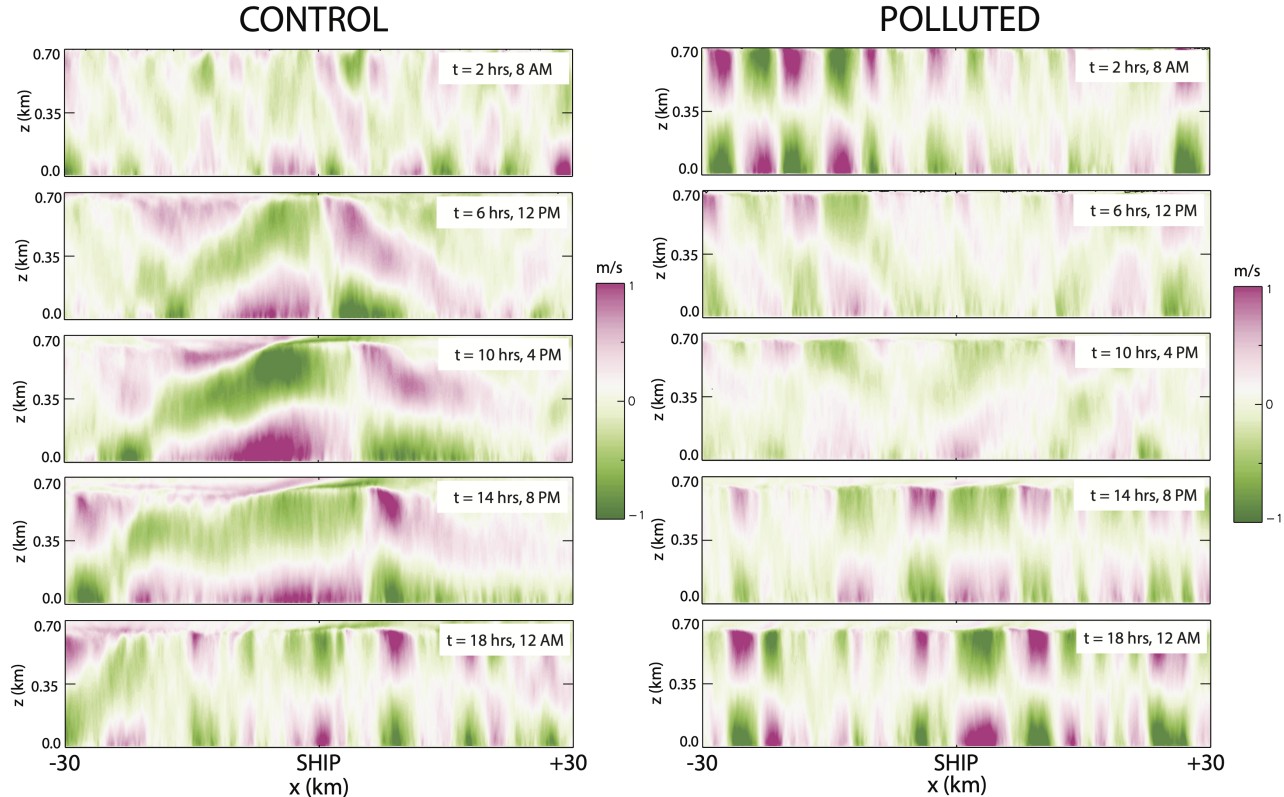

**Figure 16.** $x$-$z$ cross-sections of zonal velocity with pink going left to right and green going right to left. The CONTROL run containing a strong mesoscale circulation in the vicinity of the ship is on the left. The POLLUTED case containing no mesoscale circulation is shown on the right.

comparison to the LES shear sensitivity case widths uses domain-averaged input parameters, the modified relaxation timescale $T_m$, and a re-optimized $C_0$, named $C_m$, which accounts for the new relaxation timescale formulation.

Applying the best-performing particle model conditions (domain-averaged input parameters, $T_m$, $C_m$) to the POLLUTED case generates excessive day 1 spreading rates as the CONTROL case that $C_m$ was optimized for contained a large mesoscale

circulation that aided in plume dispersion and no mesoscale circulation exists in the POLLUTED case. Using $C_m$ in the POL-LUTED case artificially decreases dissipation although no such anisotropy is present. When $C_m$ is optimized for the POL-LUTED, non-precipitating environment $C_{np}$, we find that the particle model is again able to accurately recreate the spreading rate geometry in the LES.

The particle model is able to capture the impacts of an anisotropic, spread-accelerating mesoscale feature in precipitating

cases using only domain-averaged input parameters through a change in the free parameter, $C_0$. Using $C_m = 0.15$ for precipitating environments that behave anisotropically during the daytime and $C_{np} = 0.50$ for non-precipitating environments is one potential way of easily dealing with anisotropic drift without the addition of an extra term that would require information not



available to the subgrid parameterization, such as the spatial gradient of variance. The Langevin particle model is able to represent spreading rates better than traditional methods of constant Gaussian diffusion, all while using domain-averaged turbulence
statistics. Considering that domain-averaged information from the turbulence parameterization within a climate grid box is the only available input into a would-be subgrid particle model, the particle model performance in this study is promising, suggesting such an approach may be a viable method of cheaply and accurately modeling plume spreading in different environments. Having access to subgrid horizontal variances to calculate $T_m$ within a GCM grid box would require a higher-order closure such as CLUBB (Larson and Golaz, 2005; Guo et al., 2015) or perhaps an additional parameterization based on resolved wind
shear and boundary-layer-integrated radiative cooling.

*Code and data availability.*  The LES case setup and forcing files for the shear sensitivity tests and the non-precipitating case can be found on github at https://github.com/lmcmichael/S12_CGILS_LES_forcing (https://zenodo.org/doi/10.5281/zenodo.10557703). Matlab and IDL code to perform the Guassian curve fitting procedure, the particle model code (PM-ABL v1.0), calculation of input parameters from LES output, routines for calculating boundary-layer averaged aerosol from 3-D LES output, figure plotting procedures, and LES input parameters are
available at https://github.com/lmcmichael/ParticleModel (https://zenodo.org/doi/10.5281/zenodo.10557564). UW-SAM source code (SAM v6.10.9), additional LES source code to calculate ship track conditional averages and add momentum hyperdiffusion in SAM, and a workflow document is available here: https://github.com/lmcmichael/SAM_SHIP_TRACK_STATS/ (https://zenodo.org/doi/10.5281/zenodo.10557826). High-resolution, 3-D LES output is available upon request.

*Author contributions.*  All coauthors contributed to the conceptualization of the LES-informed particle model, with RW and LP guiding the
project initiatives and providing funding. MS and LM developed particle model code and performed simulations. LM and PB designed and carried out LES simulations. LM and MS prepared the manuscript. All coauthors took part in editing the manuscript and regular discussions regarding methods and results.

*Competing interests.*  The authors declare that no competing interests are present.

*Acknowledgements.*  This paper describes objective technical results and analysis. Any subjective views or opinions that might be expressed in
the paper do not necessarily represent the views of the U.S. Department of Energy or the United States Government. This work was supported by the Laboratory Directed Research and Development program at Sandia National Laboratories, a multimission laboratory managed and operated by National Technology and Engineering Solutions of Sandia, LLC, a wholly-owned subsidiary of Honeywell International, Inc., for both the U.S. Department of Energy's National Nuclear Security Administration under contract DE-NA0003525. This material is based in part upon work supported by the National Science Foundation under Grant No. AGS-1912130 (PB). Special thanks to Je-Yun Chun for
providing LES forcing and initialization profiles for the CONTROL and POLLUTED cases.





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
