# Peer review of "Exploring ship track spreading rates with a physics-informed Langevin particle parameterization"

_EGUsphere, 2024_

## Referee Comment (RC1)

**Comments to the manuscript with title "Exploring ship track spreading rates with a physics-informed Langevin particle parameterization" by McMichael et. al.**

May 4, 2024

**General comments**

This manuscript investigates the aerosol spread rate from a point source using a "Lagrangian particle model governed by a Langevin stochastic differential equation to create a simplified framework for predicting the rate of spreading from a ship-injected aerosol plume in sheared, precipitating, and non-precipitating boundary layers". The authors showed that "the stochastic particle-velocity representation can reasonably reproduce spreading rates in sheared, precipitating, and non-precipitating cases using domain-averaged turbulent statistics from the LES". Using statistical physics to study aerosol-airflow interactions and the consequential aerosol-cloud interactions is very novel. The manuscript is also well-written. I recommend the publication of this manuscript with the following comments for the authors to consider.

My main conceptual comment is the scale problem. It is surprising to see that using *domain-averaged* turbulent statistics from the LES as input, the stochastic model can somehow reproduce the LES spreading rate. This is because the aerosols as tracers interact with turbulence below the Kolmogorov scales. How can the *domain-averaged* turbulent statistics that filter out the native small turbulence scales transport aerosols?

**Specific comments**

- The numerical diffusion term is not included in Eq.1. How to deal with the numerical stability without the numerical diffusion term for the continuity equation, which is a well-known issue in many applications?

- Do we expect a $-5/3$ power law for the LWP spectra? If so, is it related to the turbulence energy spectra? How to explain the deviation from the $-5/3$ power law in Fig.1(b). In addition, the LWP spectra appear to be $\Delta x$ independent if I am not mistaken. What is the reason behind this?

- Taking the $\Delta x = 50$m-LES as a reference, the $\Delta x = 50$m-LES-hyperdiffusion produces about two times larger values of LWP (Fig.2a) and smaller $z_{inv}$ (Fig.2d). However, it produces boundary-layer-averaged aerosol concentration well and $R_{sfc}$ relatively well. This indicates the hyperdiffusion contributes more to the microphysical processes than to the macrophysical ones. What is the physical explanation of this observation?

- The LWP from the weak-shear LES exhibits filament structure compared to the control and strong-shear simulations in Fig.5. Is this because of the competition between the buoyancy force and shear (Richardson number)?

- It is interesting that the spatial plume evolution determines the spatial morphology of the surface precipitation rate, which should be taken into account for modelling shiptracks. Would this be one of the highlights of this study as well?

- The PM width differs the most to the LES width for the strong shear case (Fig.12 and Fig.14b). Is this because the Langevin equation can not represent turbulence well at strong shear?

- Why are the time evolution of TKE from the LES and PM so different for the control simulation in Fig.13?

---

## Author Comment (AC1)

**Response to Reviewer 1:**

**Comments to the manuscript with title "Exploring ship track spreading rates with a physics-informed Langevin particle parameterization" by McMichael et. al.**

**General comments**

This manuscript investigates the aerosol spread rate from a point source using a "Lagrangian particle model governed by a Langevin stochastic differential equation to create a simplified framework for predicting the rate of spreading from a ship-injected aerosol plume in sheared, precipitating, and non-precipitating boundary layers". The authors showed that "the stochastic particle-velocity representation can reasonably reproduce spreading rates in sheared, precipitating, and non-precipitating cases using domain-averaged turbulent statistics from the LES". Using statistical physics to study aerosol-airflow interactions and the consequential aerosol-cloud interactions is very novel. The manuscript is also well-written. I recommend the publication of this manuscript with the following comments for the authors to consider.

My main conceptual comment is the scale problem. It is surprising to see that using domain-averaged turbulent statistics from the LES as input, the stochastic model can somehow reproduce the LES spreading rate. This is because the aerosols as tracers interact with turbulence below the Kolmogorov scales. How can the domain-averaged turbulent statistics that filter out the native small turbulence scales transport aerosols?

Thank you for your time and comments. We are studying the spreading of a local aerosol source. Throughout its lifetime, that plume is mixed into the surrounding air of low aerosol concentration by small-scale eddies. However, the plume is observed to spread by approximately three boundary layer depths per hour. This spreading occurs, in part, through larger-scale eddies stretching/shearing the plume and transporting the high-aerosol parcels into the surrounding low-aerosol region. While mixing by small-scale eddies will, in reality, be required to homogenize the aerosol concentration locally, we chose to include only the transport by large-scale eddies in our stochastic model for simplicity. If the larger features are responsible for most the transport/spreading, it then seems logically feasible that domain-averaged quantities may be able to capture the general evolution of plume spreading, and perhaps, one of the more important components to capture is the daytime increase in the relaxation timescale likely associated with decoupling. We have slightly altered several references to "mixing" in the manuscript, emphasizing that larger eddies will transport/spread aerosol rather than mix/homogenize the aerosols (paragraph starting on line 439):

"It is then immediately obvious that the spreading of the aerosol is not solely related to mixing done by the smallest scales. As an alternative calculation, D can be estimated as the product of a characteristic zonal eddy velocity (uc) and a characteristic length scale (l) on which the transport occurs (D = ucl). In the CONTROL run, the characteristic zonal eddy velocity is ≈ 0.3 m s−1 and the characteristic length scale may potentially be driven by mesoscale eddies which are initially

≈ 8 km wide. The resulting constant Gaussian diffusion equation estimates a width of 26.3 km at hour 10, which is much closer to the LES plume width (Figure 10). While assuming the larger scales are doing a majority of the transport related to the ship track spreading does improve the simple Gaussian diffusion model performance,…"

We agree that the performance of domain-averaged turbulent statistics is surprising, but also encouraging given that subgrid plume properties are unavailable in most cases. It's also important to note that while domain-averaged turbulent statistics can reproduce spreading rates with reasonable accuracy, the in-plume statistics more realistically capture the geometry of spreading in the first 15 hours, with more linear spreading and a pronounced inflection point during the evening. It is only during the overnight period in which the in-plume statistics suffer, particularly in the STRONG case. Also, the in-plume statistics are directly sampling the region in which a local mesoscale circulation exists (for the precipitating cases), likely resulting in the better initial performance, but once multiple cells develop within the plume region the daytime relationship between the optimized constant (C_m) and the dissipation seems to break down entirely (in STRONG case). This is briefly discussed in the conclusions on lines 537-539. We have added additional explanation and rearranged the text to expound on the reasoning behind the in-plume statistics failure after sunset on line 535:

"In-plume turbulent statistics perform better than domain-averaged quantities during the first 15 hours after injection, but as nocturnal turbulence disrupts the mesoscale circulation the daytime relationship between the plume-optimized turbulence constant ($C_m$) and the dissipation rate breaks down and results in larger errors thereafter. As the sun sets, the domain-averaged statistics continue to represent spreading rates well during the night and into day 2, potentially as a result of the domain-averaged $C_m$ being less sensitive to the termination of the mesoscale circulation."

Specific comments

• The numerical diffusion term is not included in Eq.1. How to deal with the numerical stability without the numerical diffusion term for the continuity equation, which is a well-known issue in many applications?

Our goal with Equation 1 was to start from first principles. If we were to discretize the conservation equation a numerical diffusion term would be introduced, but numerical diffusion is minimized in the LES by using a 5th-order advection scheme. In its current form, Eq. 1 represents a continuous conservation equation with no unphysical numerical diffusion. The main purpose of Section 2.1 is to provide a brief and vastly simplified review of the physics governing large-eddy simulation. We slightly modified the following sentence beginning on line 104 to clarify the purpose of Section 2.1:

"In the following sections, we will lay out the equations that govern our atmospheric plume model. We will begin with the Eulerian formulation representative of the LES framework and

from there, work towards the Lagrangian formulation that corresponds to the numerical particle model we introduce in Section 2.3."

- Do we expect a −5/3 power law for the LWP spectra? If so, is it related to the turbulence energy spectra? How to explain the deviation from the −5/3 power law in Fig.1(b). In addition, the LWP spectra appear to be $\Delta x$ independent if I am not mistaken. What is the reason behind this?

The near -5/3 power law for liquid water path power spectra was from satellite observations of northeast Pacific stratocumulus in Wood and Hartmann (2006) and also seen in a few other studies (Catalan and Snider, 1989; Wood and Taylor, 2001). The -5/3 slope is only expected at high frequencies (> 0.1 km⁻¹), which is in general agreement with the LWP spectra from the LES, although the -5/3 dashed line stretching the length of the x-axis in Figure 1b is confusing and has been altered to be consistent with the observations. It does appear that there is $\Delta x$ independence in the spectra and the main rationale of showing the LWP power spectra was to illustrate that if one was to only examine the variance structure of LWP at different grid spacing the conclusion would be that 200 m is sufficient; However, the analysis of rain rates, boundary-layer aerosol, and boundary-layer depth tells a much different story.

[Figure]

FIG: altered panel in Figure 1.

- Taking the $\Delta x$ = 50m-LES as a reference, the $\Delta x$ = 200m-LES-hyperdiffusion produces about two times larger values of LWP (Fig.2a) and smaller zinv (Fig.2d). However, it produces boundarylayer-averaged aerosol concentration well and Rsfc relatively well. This indicates the hyperdiffusion contributes more to the microphysical processes than to the macrophysical ones. What is the physical explanation of this observation?

This was an unexpected result and the mechanisms are not fully understood. Taken alone, the near double in LWP and in-line aerosol concentrations would be expected to produce much stronger rain rates than the 50m reference case. We mentioned in the manuscript that the inability of the 200m run to capture the rain rate seems unrelated to entrainment, but did not speculate on the potential reasons for the inability of the 200 m run to produce high enough rain rates. It's

possible that at 200m the spatial organization is disrupted/under-resolved to a point where the structure of the precipitating cells is materially different. Note that the 200m run has the smallest peak in the LWP spectrum at 8km (Figure 1b). It's possible that the LWP field is more homogeneous with the thicker parts of the cloud not generating as much precipitation as in the finer grid spacing runs.

- The LWP from the weak-shear LES exhibits filament structure compared to the control and strong-shear simulations in Fig.5. Is this because of the competition between the buoyancy force and shear (Richardson number)?

It's difficult to pin down the exact cause of the filament-like structure, but it appears one of the main differences between the no shear run and the others is the much lower entrainment rate, which is likely due to less shear-driven mixing near cloud top (locally, Ri $\gg$ 1). The reduced mixing maintains lower boundary-layer aerosol concentrations and continued aerosol scavenging from ongoing precipitation which both promote larger precipitation rates. The larger precipitation rates are likely driving a faster transition to open-cellular convection as cold pools merge and the narrow cloud filaments form where cold pool mergers occur.

- It is interesting that the spatial plume evolution determines the spatial morphology of the surface precipitation rate, which should be taken into account for modeling ship tracks. Would this be one of the highlights of this study as well?

The down-shear enhancement of surface precipitation in the CONTROL and STRONG cases is notable and a signal that persists for the entire simulation. As far as we know, the down-shear precipitation enhancement has not been mentioned in previous ship track studies. We have added a few more words near line 338 to point out the interesting result in Figure 7.

"Local precipitation enhancement occurs on the down-shear side of the plume edge in the CONTROL and STRONG cases, becoming especially prominent during the second daytime period (Figure 7)."

- The PM width differs the most to the LES width for the strong shear case (Fig.12 and Fig.14b). Is this because the Langevin equation can not represent turbulence well at strong shear?

The PM seems to perform best during the most intense in-plume zonal variance (first ~10 hours after injection) and then begins to deviate from the LES strongly near sunset, as in-plume turbulence begins to wane and relax to domain-averaged values. It is after this reduction in turbulence that it seems the relaxation timescale may be artificially high in the absence of the mesoscale circulation that existed during the daytime.

- Why are the time evolution of TKE from the LES and PM so different for the control simulation in Fig.13?

In Figure 13, all time series are from the LES. The solid lines are the domain averages and the dashed lines are the in-plume averages. The confusion is understandable given that Figure 12 used solid lines for LES and dashed lines for the PM. We changed the dashed lines to dot-dashed lines to hopefully make this distinction more clear. Clarification was added to the figure caption to emphasize that the dashed lines are in-plume quantities and solid lines are domain averages.

"Figure 13. (a) The original relaxation timescale formulation ($T\_L$) and (b) the modified version of the relaxation timescale ($T\_m$), which focuses only on the zonal variance. Solid lines correspond to domain-averaged time series and dot-dashed lines correspond to in-plume (STcloud) averaged time series."

---

## Author Comment (AC2)

**Response to Reviewer 2:**

**Comments to the manuscript with title "Exploring ship track spreading rates with a physics-informed Langevin particle parameterization" by McMichael et. al.**

**General comments**

This article presents a novel approach to modeling the spreading of aerosol plumes in a marine boundary layer capped with stratocumulus clouds. The model is inspired from the stochastic processes based approaches used in turbulent flows for representing Lagrangian trajectories. The results presented in this work will be of interest to the marine cloud brightening and ship-track community (broadly speaking - ACI community) and should be considered for publication after the comments listed below are carefully addressed.

Thank you for taking the time and effort to review our manuscript and improve the clarity of the paper as a whole.

We would first like to address a recurring comment throughout the review. The reviewer mentions several times that "effective" diffusivity could be a simple way of modeling the spread. We acknowledge that given LES spreading rates, the effective diffusivity can be obtained; however, the effective diffusivity would be a function of time and dependent on environmental conditions. To learn effective diffusivities under the entire range of atmospheric turbulence properties possible in the atmosphere would require an enormous LES library of high-resolution plume spreading simulations (which require very wide domains and multiple days of simulation, typically costing 200,000 core hours or more per simulation). The computational cost combined with the range of environments necessary makes such an exercise cost prohibitive at the current time, but at some point in the future such a machine learning study could be carried out. For the purposes of this study, the hope was that a properly tuned Langevin equation could capture the physics of many different plume spreading environments all while being computationally efficient. We have added a section at the end of the conclusions section discussing the possibility of machine-learned effective diffusivities with a future large library of ship-plume LESs.

"It is also worth noting that given a sufficiently large library of high-resolution LES, it may be possible to machine learn a time-dependent $\mathscr{D}$ in (34); however, the computational cost of multi-day ship track simulations currently precludes the creation of such a library."

Comments:
1. Can you explain the physical meaning behind the terms in the GLM/SLM model?

The physical meaning behind the terms are perhaps most easily viewed as a scale separation problem. The deterministic drift term in (23) can be conceptualized as representing larger-scale turbulent structures that are governing particle trajectories through changes in the relaxation timescale. The Brownian motion term in (23) can be thought of as representing the smallest

eddies in the flow and attempts to account for the random, small-scale turbulent fluctuations. We have added some text on line 210:

"Conceptually, the deterministic drift term is related to the representation of larger-scale flow features, while the Brownian motion term attempts to represent the random, smaller-scale turbulent fluctuations."

Regarding the GLM, two of the terms are essentially the same as in the SLM, up to the definition of the model constants $G_{ij}$ and $C_0$, though the differing term

$$-\frac{1}{\rho}\frac{\partial P}{\partial x_i}dt,$$

is described by Pope as the "drift term in the mean pressure gradient". In more descriptive terms, it functions as an advection-like motion that is driven by the mean pressure gradient. We are of the opinion that including this description of the extra term in the GLM would add extra "noise" without adding much to the understanding of the particle model we employ in this work. However, if we have misunderstood or the reviewer disagrees with our assessment, we are happy to add the desired level of detail.

2.  Eq 16, why would the boundary layer flow allow isotropic coeff for the drift term? Is it because you are considering only the horizontal dimensions?

Great catch! Thank you for pointing that out because that bit is poorly stated, and the boundary-layer isotropy would only explain the horizontal components. The proper description is that choosing an isotropic drift coefficient, $G_{ij}$, is a modeling choice to reduce the generalized Langevin model (GLM) to the simplified Langevin model (SLM). There are other (non-isotropic) choices for $G_{ij}$ and the corresponding model parameter $C_0$ that result in other specializations of the GLM, for instance Rotta's Model adds a cross-derivative term to $G_{ij}$ that is almost certainly non-isotropic. We have changed the text to clarify this (line 189):

"Second, we impose an isotropic weight coefficient for the drift term, namely…"

3. In eq 18, are TL and Co constants?

$C_0$ is indeed a model constant, as described above. $T_L$ however, does vary in time and is related to other variables that change with time, as given in Equations (16), (19), and (20). We have made a slight change to the introduction of $G_{ij}$, emphasizing the time-variable nature on line 180:

"… Additionally, we introduce the constant $C_0$ and a time-variable drift coefficient $G_{ij}$"

4. What are the overbar terms in Eq 23 and 14? Shouldn't those be zero?

The overbar term in those equations, $\overline{U_j^* | \mathbf{X}^*(t)}$ and the vectorized version $\overline{\mathbf{U}_t^*}$, are the mean Lagrangian velocity, with the star denoting the Lagrangian quantity and the overbar indicates a mean value. The model we consider contains nonzero mean velocity leading to those terms sticking around. If the reviewer is familiar with Pope's *Turbulent Flows* text, our model is distinct from that given in Sec. 12.3.1, Eq. (12.89) that does assume zero-mean velocity and thus lacks the overbar terms.

5. Eq 30, explain how TL and \sigma are obtained?

If the question is seeking clarity on how those parameters are chosen for a simulation that uses this model, we would clarify that, for the results of this work, we employ the boundary-layer averaged values provided by the LES model we wish to validate against. This is discussed in detail in Section 3, specifically Section 3.3. If the question refers to how those terms arrive in that equation, then we note that this equation is a forward-Euler-integrated form of the referenced Eq (23), where the terms originate in the model constants $G_{ij}$ and $C_0$.

6. Lines 280-290: Are the surface fluxes identical for all the grid and domain sizes? I am wondering about the differences in LWP/precip. and if they are related to the differences in the surface fluxes? So, is Eq. 32 required?

The surface fluxes are interactive and allowed to respond to local flow conditions. The main indication that the difference in precipitation rate is driven by entrainment and not surface fluxes is that boundary layer depth is increasing in the non-hyper diffusion runs (Figure 2d). Since the large-scale prescribed subsidence remains the same in all of the runs the increase in boundary layer depth is a result of entrainment (mass budget equation: $\dfrac{Dz_i}{Dt} = w_e + w_s$, where the material derivative of inversion height ($z_i$) is controlled by the entrainment rate ($w_e$) and large-scale subsidence ($w_s$)). We have added some clarification on line 279 to explain why the increased boundary layer depth is a result of larger entrainment rates:

"Given that large-scale prescribed vertical motion remains the same between runs, an increase in boundary layer depth in comparison to the CONTROL run indicates higher entrainment rates."

7. Line 315: the unit of shear is wrong.

The shear magnitudes in the paper are calculated as a bulk wind difference of the layer in question (top zonal velocity vector - bottom zonal velocity vector). This differs from the standard dU/dz calculation as the reviewer mentions. We have added clarification to state that we are calculating bulk wind shear on line 313:

"If the cloud base is found to be below the estimated boundary-layer height and the cloud depth exceeds 50 m we compute the boundary-layer, cloud-layer, and subcloud-layer zonal shear magnitudes as a bulk wind (vector) difference between the top and bottom of the layer."

8. Section 3.1, the forcing used for obtaining the different shear rates are not clear? Note that other readers should be able to reproduce these results. More details would be useful.

We have added the the large-scale forcing files to Github that include the wind profiles for all of the cases (https://github.com/lmcmichael/S12_CGILS_LES_forcing). Landing at reasonable post spin-up shear magnitudes is difficult to do, especially when we focused on trying to limit changing the shear near cloud top. Many different profiles were tested on small domains to create the final profiles which fall within the range of reasonable boundary-layer shear magnitudes in stratocumulus clouds.

9. Lines 380-395: I think the figures numbers are wrong. Please check.

Nice catch! There are quite a few of erroneous references to Figure 8, which should be Figure 9. This has been fixed.

10. Lines 402-405: The reasoning is very vague. You need to show that there is some difference in the energy cascade. Can you?

If the dissipation rate changes for a fixed amount of TKE this suggests that the transfer of energy from larger scales to smaller scales is happening at a different rate and thus, the energy cascade has been altered. The latter half of the sentence *"suggesting that the energy cascade at small scales is fundamentally different and the plume turbulence may be dominated by larger coherent structures that dissipate energy less efficiently, such as a mesoscale circulation driven by precipitation suppression."* is speculating as to why the energy cascade may be behaving in such a way. The large organized flow features may have large regions of relatively small velocity gradients which lessen the efficiency of dissipation. The next few sentences are also spent speculating on other mechanisms which may cause the dissipation rate difference, such as TKE transport and stratification, but the fact that the energy cascade is different is supported simply by the dissipation rate changing relative to the magnitude of the TKE. We have reorganized the following section to separate the statement on the changing energy cascade from the speculation of why the dissipation rates are different:

"Although the TKE in the ship track region is larger than the domain-averaged TKE, the dissipation rate is smaller during the evening of day 1 (Figure 9d,e), suggesting that the energy cascade at small scales is fundamentally different. There are several factors which may contribute to less efficient dissipation and an altered energy cascade. We speculate that the mesoscale circulation which develops in precipitating cases may cause reduced dissipation rates as a result of changing flow geometry, as larger coherent structures within the circulation have

smaller internal velocity gradients. The dissipation rate is also dependent on stability, as more stable environments dissipate energy to smaller scales at a slower rate. In comparison to the non-ship track region, the ship track region is more stable and experiences larger negative buoyancy fluxes near cloud base and stronger daytime decoupling, which also leads to longer ship-track relaxation timescales as a result of less efficient dissipation (smaller dissipation rate) given a fixed amount of TKE. Additionally, organized horizontal TKE transport in the ship region region may be causing the TKE to be dissipated in the non-ship region. The relative importance of each of these dissipation-modulating mechanisms is currently unknown."

11. Same for line 406.

Hopefully, we have addressed the reviewer's comment in the previous response.

12. Line 408: what do you mean by efficiency of dissipation?

It refers to how effectively TKE is converted to thermal energy by viscous forces. We have altered the following sentences slightly:

"The dissipation rate is also dependent on stability, as more stable environments dissipate energy to smaller scales at a slower rate. In comparison to the non-ship track region, the ship track region is more stable and experiences larger negative buoyancy fluxes near cloud base and stronger daytime decoupling, which also leads to longer ship-track relaxation timescales as a result of less efficient dissipation (smaller dissipation rate) given a fixed amount of TKE."

13. Line 439: why is sgs diffusivity the relevant parameter? Can't you find an effective diffusivity from the resolved and SGS stresses? That would be the appropriate parameter.

Our original assumption was that the mixing of the plume may have been achieved by small scales near the plume edges. Using SGS diffusivity output from the LES 1.5-order TKE closure allowed us to discount the possibility of small-scale-driven diffusion and shifted our focus to larger turbulent structures. As for the effective diffusivity, this could be backed out from the plume spreading data as mentioned previously, although, it does not solve overarching problem of being able to represent plume spreading in a wide range of turbulent environments.

14. Line 448: which part has the superlinear growth? Can you show it in the figure? I dont find any superlinear growth. Can you clarify? And what about numerical diffusion? How do you account for that?

The blue outlined region is where the enhanced growth rate occurs in the CONTROL run. It was growing at ~2 km/hr for the first 9 hours (dashed line is 2 km/hr growth rate), but around hour 10 an inflection upward occurs and the growth rate diverges from 2 km/hr. This is a subtle uptick and hard to visibly detect in the zoomed out figure in the paper so we have added additional detail and dropped the "superlinear" statement:

"…it fails to capture the accelerating growth (growth rates grow from 2 km/hr in the first 9 hours, to near 3 km/hr during hours 9-13) in the first 15 hours after injection in the CONTROL run (Figure 10)."

[Figure]

FIG: Dashed line is 2 km/hr growth rate. Bold black line in the LES plume width for the first 15 hours. The spread deviates from 2 km/hr growth near hour 10 before slowing near hour 13.

As for numerical diffusion, the LES uses a high-order (5th) advection scheme along with a flux limiter to accurately advect scalars in the presence of sharp gradients (Yamaguchi et al., 2011).

15. How do your growth rates compare against other studies?

Satellite observations of plume growth rates have generally been ~2 km/hr (Heffter, 1965; Durkee et al., 2000), which seem reasonably close to the CGILS case examined in this paper (generally 1-3 km/hr). In Prabhakaran et al. (2024) and some of our recent simulations, deeper boundary layer cases associated with a stratocumulus-to-cumulus transition (SCT) (Idealized setup laid out in Sandu and Stevens, 2011) can have considerably faster spreading rates (> 5 km/hr) than the shallow boundary layer simulations in this paper.

16. Fig 10, what about the diffusive model with diffusivity changing with time? And can you plot the diffusive model for each case and not just the control case (maybe in Fig 12). This is required to check if the particle model is actually better.

A changing "effective" diffusivity in time is the end goal of the Langevin particle model. Without some relationship to the physics, it's not clear how to change the diffusivity in time, other than to match the LES growth curves. With a large enough library of LES plume spreading rates it would be possible to learn the time-dependent effective diffusivity; However, in this work, we wish to create a less data-dependent framework to predict spreading rates given the high computational cost of plume spreading simulations (100s of thousands of core hours).

In the CONTROL case, the characteristic length scale was chosen using the characteristic mesoscale cell size at initialization (~8 km) as determined from LWP power spectra. Instead, in the following plots we will pick characteristic cell sizes that best capture the spreading rate. Here are the plots of the diffusive model for each case:

[Figure]

FIG: The best fit Gaussian diffusion curve (dot-dashed thin line) plotted over the LES width (black line) and particle model ensemble mean result (dashed blue line). Even choosing

diffusivity values (not related to large-scale features as in the CONTROL case) that fit the LES width best, the particle model performs better.

17. Why is the plume growth rate lower in the night time? Was it seen in other studies as well?

This seems to be case dependent. In the S12 CGILS case examined in this paper, the shallow boundary layer is able to recouple during the nighttime hours, causing the relaxation timescale to converge to similar values for all runs. Additionally, the variance is relatively constant throughout the period, causing the overnight spreading rates to decrease as a result of lower relaxation timescales (particle velocity is more aggressively nudged back to the mean wind). In Prabhakaran et al. (2024) and our recent plume spreading simulations, plume spreading is rapid overnight as the decoupled boundary layer continues to deepen, variance increases, and the relaxation timescale continues to increase. A comparison of the Sandu and Stevens (2011) SCT case spreading rate with the spreading rates in the current manuscript is shown below:

[Figure]

FIG: Plume widths for all of the cases in this paper, along with an additional idealized SCT case setup from Sandu and Stevens (2011) (red line). Also, important to note that a square root growth curve would result in extremely poor performance in the SCT case.

We have added a comment in the conclusions to emphasize that spreading rates are not always expected to be slower overnight and that future studies will focus on cases with drastically different timescale and variance parameters.

"While we have examined a broad range of shear magnitudes in this paper, only a small portion of the variance/relaxation timescale parameter space has been explored. Deeper boundary-layer cases associated with stratocumulus-to-cumulus transitions can have accelerating spreading rates overnight \citep{Prabhakaran2024EffectsOI} with spreading rates in excess of 4 km/hr. Future plans are to apply the particle model to a much broader range of conditions."

18. Sec 4.2, how is TL for the particle model determined? Is it time varying? I see that Co is from a least sq. fit. Why not make it time varying as well?

TL for the particle model is time varying and is determined in several different ways in the paper. TL is calculated the same as Tm (35), but instead of using the horizontal variance, the entire TKE (k) was used. We have rearranged (18) in terms of $T\_L$ to hopefully make this more clear. Varying $C\_0$ in time is similar to using the effective diffusivity approach the reviewer mentioned previously. This type of parameter fitting would have to be done on a large library of LES and is, therefore, not desirable for our current goals with the particle model.

19. Lines 474-479: Please show these.

We don't believe it is necessary to show the lower particle number results in the manuscript, but 3 separate realizations of 1,000 and 400 particle simulations are shown below. Even at 400 particles, the solution remains quite stable

[Figure]

FIG: ensemble particle results for varying total particle number: (top row) 50 ensemble members, 50 particles each and (bottom row) 20 ensemble members, 20 particles each.

20. The authors have articulated in the article that the particle based estimates are better than the Gaussian-diffusion based estimates. But the Ornstein-Uhlenbeck process used here for obtaining the velocity field is a Gaussian diffusive process (i.e., random walk based on a Gaussian distribution). Then why would the two give different results? My impression was that it's a question of having the right effective diffusivity. A discussion related to this would be useful.

The foundational principle of a Langevin equation is that it contains both contributions from both deterministic and random components. The addition of the drift term makes the Langevin approach distinctly different than a purely random diffusive process. A discussion on effective diffusivity and the motivation for the particle model has been added to the conclusion (see first major comment).

21. References: I think the reference to Prabhakaran et al 2024 (ACP) is more relevant than 2023 (JAS).

That's correct. Thank you for catching that.

22. Lines 45-50: Do the other recent LES studies agree with the conclusions of Chun et al 2023?

In Chun et al. (2023), the sensitivity to injection rate was not explored thoroughly, but the results generally align with the stratocumulus transition cases examined in Prabhakaran et al. (2024), with no substantial cloud darkening occurring in any of the simulations over the entire 4-day simulations (polluted or clean) and the largest cloud radiative effects realized through a change in cloud fraction associated with precipitation suppression in the clean cases.

23. Lines 78-90: parts of this paragraph doesn't seem like introductory text. This could go into abstract or in sec. 4.2. Sec 4.2 would be preferred.

This section is a more detailed version of the abstract (although, a restatement) and serves to set up the approach/methods for the rest of the paper and we believe it fits well with the current placement.